JCB Journal of Cell Biology

# Translation of unspliced retroviral genomic RNA in the host cell is regulated in both space and time

Felipe Leon-Diaz[1], Célia Chamontin[1], Sébastien Lainé[1], Marius Socol[1], Edouard Bertrand[2], and Marylène Mougel[1]

Retroviruses carry a genomic intron-containing RNA with a long structured 5′-untranslated region, which acts either as a genome encapsidated in the viral progeny or as an mRNA encoding the key structural protein, Gag. We developed a single-molecule microscopy approach to simultaneously visualize the viral mRNA and the nascent Gag protein during translation directly in the cell. We found that a minority of the RNA molecules serve as mRNA and that they are translated in a fast and efficient process. Surprisingly, viral polysomes were also observed at the cell periphery, indicating that translation is regulated in both space and time. Virus translation near the plasma membrane may benefit from reduced competition for ribosomes with most cellular cytoplasmic mRNAs. In addition, local and efficient translation must spare energy to produce Gag proteins, where they accumulate to assemble new viral particles, potentially allowing the virus to evade the host's antiviral defenses.

## Introduction

Retroviruses are a class of enveloped viruses with single-stranded RNA genomes (gRNA). They have long captivated scientists due to their remarkable ability to integrate their genetic material into the host cell's genome, with profound implications for both biological research and human health. The murine leukemia virus (MLV), among the first retroviruses identified in mice in the early 1950s, stands as a model system for basic virology and cancer research. MLV has significantly influenced our understanding of retroviral replication, the dynamics of retrovirus–host interactions, oncogenesis, and the development of gene therapy vectors (Cavazzana-Calvo and Fischer, 2007; Dudley, 2003; Fan, 1997; Mikkers et al., 2002; Suzuki et al., 2002).

Like all retroviruses, the MLV replication cycle is composed of early and late phases. Briefly, early phases include virus entry, RT of gRNA into DNA, and subsequent integration of the latter into the host cell chromosomes. Transcription of the proviral DNA into a single full-length (FL) transcript (called FL RNA or gRNA) initiates the late steps. The latter are of interest to RNA biologists, as FL RNA displays atypical and complex fates governed by poorly understood mechanisms. First, FL RNA splicing is incomplete and finely regulated: some molecules are fully or alternatively spliced (Déjardin et al., 2000) to generate mRNAs encoding envelope and regulatory proteins, while others escape splicing and, despite the persistence of introns, are nevertheless exported from the nucleus (Hoshi et al., 2002; Houzet et al., 2003; Yap et al., 2000). Once in the cytoplasm, FL RNA has the following two destinations and functions: either FL RNA is handled by cellular ribosomes to serve as mRNA for the synthesis of the structural (Gag) and enzymatic (Pol) polyproteins, or it travels to the plasma membrane (PM) to act as a genome that is packaged as dimers in nascent viruses (Dubois et al., 2018). Our recent studies reveal the use of two TAP (NXF1)- and CRM1-dependent export pathways to exit FL RNA from the nucleus (for review [Pessel-Vivares et al., 2015] and references therein) and show that the TAP pathway targets FL RNA to the translation machinery, while CRM1 export marks FL RNA for encapsidation during viral assembly at the cell periphery (Mougel et al., 2020).

MLV's genetic minimalism renders it an ideal model to dissect the roles of individual genes and proteins in the virus life cycle. Indeed, MLV has long been considered as a prototypical simple retrovirus, with just three major genes, encoding the structural proteins (Gag) that form the viral core, the enzymatic machinery (Pol) required for maturation, integration, and RT, and the envelope proteins (Env) that mediate host cell entry (Fig. 1 A). However, this notion of genomic simplicity needs to be weighed against the fact that MLV also encodes a glycosylated Gag protein, the ~10 kDa extended glycoGag (gGag) (Prats et al., 1989) and also a transcriptional regulatory protein, p50, derived from alternative splicing (Akkawi et al., 2023). These two accessory proteins, gGag and p50, prevent the cellular protein APOBEC3 from blocking MLV infection (Rosales Gerpe et al., 2015; Stavrou et al., 2013; Zhao et al., 2020). The gGag protein also counteracts SERINC5 antiviral activity by promoting its relocalization to the endosomal compartment (Diehl et al., 2021).

[1]Team R2D2: Retroviral RNA Dynamics and Delivery, IRIM, UMR9004, CNRS, University of Montpellier, Montpellier, France;   [2]IGH UMR 9002 CNRS, Université de Montpellier, Montpellier, France.

Correspondence to Marylène Mougel: marylene.mougel@irim.cnrs.fr.



Figure 1. **Visualization of viral FL RNA translation by dual RNA-Gag labeling. (A)** Schematic representation of the MLV construct. MLV-ST-MS2 contains an in-frame insertion of the 24XST in Gag and the 24XMS2 in Pol. Ψ refers to the RNA encapsidation signal, splice donor (SD, SD') and acceptor (SA) sites are shown, as well as the start codons for Gag (ATG) and for gGag (CTG). **(B)** Representative images of a GFP/RFP cell expressing WT MLV-ST-MS2 acquired on a widefield microscope. Merge and maximum intensity projection (MIP) of several z-stacks are presented on the left. A cell treated with puromycin is shown in the lower panels. In the zoom insets, red arrows indicate non-translating RNA, yellow arrows indicate translating RNA, and green arrows indicate mature Gag. Scale bars are 10 and 1 μm for zoom insets. **(C)** Quantification of MS2 and ST spots per cell detected with Imaris in the presence or absence of puromycin. **(D)** Proportion of MS2 spots colocated with ST spots per cell. For all graphs, the mean and SEM are shown with $n$ = 38 cells. The significance of differences was assessed using a nonparametric test (Mann–Whitney) (ns = nonsignificant, ****P ≤ 0.0001).

Here, we focus on the synthesis of the main structural Gag protein, which directs virus assembly and FL RNA packaging (reviewed in Bernacchi [2022]; Darlix et al. [2014]; Rein et al. [2011]). In the cell, Gag is synthesized as a polyprotein (65 KDa), which is then cleaved in virions by the viral protease into several individual proteins, including matrix, p12, capsid, and nucleocapsid (NC) (Fig. 1 A). Gag is the most abundant protein, since thousands of copies must assemble to form a virion (Lavado-García et al., 2021). Translation of FL RNA into Gag polyprotein is an intricate and tightly regulated process, which must produce sufficient viral components to ensure virus production (Odawara et al., 1998). Gag synthesis relies on cap structure at the 5′ end of FL RNA and ribosome scanning (see review [Guerrero et al., 2015]). The ribosome scans along a particularly long 5′ UTR of FL RNA until it encounters a conventional translation initiation codon, AUG, which marks the start of protein synthesis, just like cellular mRNAs. However, an upstream CUG in 5′ UTR can also serve as an alternative start codon, in the same open reading frame as the AUG$_{gag}$, producing the glyco-polyprotein gGag (Edwards and Fan, 1979; Irigoyen et al., 2018; Prats et al., 1989). In addition, putative internal ribosome entry sites were reported in the 5′ UTR (Berlioz and Darlix, 1995; López-Lastra et al., 1997), as observed in other retroviruses (Balvay et al., 2009). In both cases (CUG or internal ribosome entry sites initiation), the translated proteins are barely detectable by standard western blot analysis. Translation termination occurs at the UAG stop codon at the junction of the gag and the pol genes, which is bypassed once every 10 times by a read through mechanism to produce the Gag-Pol fusion polyprotein (Jacks, 1990; Yoshinaka et al., 1985). A recent study on ribosome profiling of MLV indicated that ∼7% of ribosomes undergo read through to access the pol gene (Irigoyen et al., 2018). To our knowledge, MLV translation has not been fully explored, especially at the scale of a single cell, at single RNA molecule resolution, or in real time.

Recent advances in high-resolution and live microscopy have made it possible to study translation directly in the cell. A recent breakthrough in the field was brought by the SunTag (ST) technology that monitors the translation of single-mRNA molecules by rapidly detecting nascent proteins (for review [Basyuk et al., 2021; Schmidt et al., 2020] and references herein). ST approach enables the visualization of the nascent peptide in cell at a single-peptide molecule scale. The protein tag consists of multiple copies of a small peptide sequence, typically 24 tandem copies of ST peptide bound with high affinity and specificity by a specific antibody (GCN4) fragment single-chain variable fragment (scFv) fused to a superfolder GFP (sfGFP), including a small solubility tag GB1 (scFv-GCN4-sfGFP-GB1, hereafter referred to as scFv-GFP) (Tanenbaum et al., 2014). To study MLV translation, we combined the ST tool with MS2 RNA imaging to visualize FL RNA and the nascent peptide (Gag) simultaneously. The 24 MS2 hairpin repeats allow RNA visualization with MS2 coat protein, MCP, fused to red fluorescence protein, RFP (MCP-RFP) (Bertrand et al., 1998; Chartrand et al., 2000; Darzacq et al., 2007). We adapted these two single-molecule imaging approaches to the study of MLV translation. Experiments were conducted with a Gag protein lacking its NC domain (MLV ΔNC),

which specifically binds FL RNA, to prevent FL RNA–Gag interaction during virus assembly. Thus, colocalization of cytoplasmic MCP-RFP (bound to FL RNA) and scFv-GFP (bound to nascent Gag) signals correspond to polysomes and not to assembly events. This approach allowed direct measurements of the proportion of translating FL RNA, the efficiency of the translation mechanism, and the intracellular localization of MLV translation sites.

## Results

### Visualization and quantification of translating FL RNA in cell

One of the key issues in our imaging approaches was the choice of tag positions within the whole MLV molecular clone to affect MLV replication as little as possible. 24 copies of ST (24XST) were inserted in frame inside p12 domain of Gag to adapt the ST tool to MLV translation (Fig. 1 A). This position corresponds to the insertion site of the GFP reporter gene used by C. Baum's team, who showed that GFP-Gag remains functional (Voelkel et al., 2010). However, the GFP-Gag fusion was not appropriate for the present study, as GFP signals are too weak to visualize single polysome in real time. The choice of RNA tag position was also critical, as FL RNA functional sequences often overlap. In our previous works, we have already inserted 24XMS2 in the intronic region of the MLV pol gene to specifically visualize FL RNA (and not spliced RNAs) during packaging (Mougel et al., 2020). Thus, we tagged MLV with both 24XST and 24XMS2 repeats (MLV-ST-MS2). When the latter was expressed in cells, some virions were still detected in culture medium, indicating that 24XST was tolerated for complete Gag synthesis (of the expected size) and for subsequent virion assembly and release (Fig. S1). The particularity of our approach was to study translation in a viral context that preserves as much as possible the different steps in which FL RNA and Gag are engaged upon viral replication (LTR-dependent transcription, translation, and virus assembly). For this reason, unlike other translation studies using ST methodology (Balme et al., 2016; Chen et al., 2020; Pichon et al., 2016; Tanenbaum et al., 2014; Wang et al., 2016), the protein produced (here Gag) was not fused to a degron nor deliberately targeted to a particular cellular compartment after its synthesis completion. Also, the MCP-RFP and scFv-GFP, both harboring a nuclear localization sequence (NLS), were cloned into non-retroviral vectors to avoid any potential interference with MLV expression (Fig. S2 A), and then stably co-transfected into murine NIH3T3 cells (see Materials and methods). As expected, scFv-GFP and MCP-RFP, both fused to NLS, localized to the nucleus (Fig. S2 B), decreasing background signals in cytoplasm. Transient expression of the FL RNA in this stable cell line resulted in the appearance of bright green puncta in the cytoplasm corresponding to Gag-24XST peptides (Fig. 1 B). As the ST was inserted in the middle of the gag gene, the green signals may be underestimated, but only slightly, as the spots are already detectable at 5xST (Boersma et al., 2020). If the green spots correspond to active translation, then mRNA and nascent polypeptide chain should colocalize (Fig. 1 B). After image analysis (Fig. S2 C), all red and

green signals were detected by using Imaris software and quantified (Fig. 1 C) to determine the level of translating FL RNA per cell (Fig. 1 D):

Fraction of translating FL RNAs =

$$\frac{\text{\# of colocalized red and green dots}}{\text{\# of red dots}}$$

A correlation analysis between the percentage of translating RNA (colocalized red–green dots) and the total number of red dots in the cell shows that the percentage of colocalization does not depend on the density of red dots per cell (Fig. S2 D). To confirm that colocalized dots were bona fide translation sites, cells were treated with puromycin, an inhibitor of translation that dissociates ribosomes from RNA (Blobel and Sabatini, 1971). After a few minutes of treatment, there was indeed a decreased number of green spots, but no change was observed in the number of red spots. Consequently, there was a reduction in the level of colocalized spots (Fig. 1, B–D). However, 7% of colocalized dots persisted after puromycin treatment, contrasting reports in other studies (Morisaki et al., 2016; Pichon et al., 2016; Wang et al., 2016; Wu et al., 2016; Yan et al., 2016). As expected, a translation inhibitor as the cycloheximide, which stalls the ribosome on mRNA, did not alter the proportion of translating FL RNA (Fig. S2 E).

The role of the NC domain of Gag in the recognition of the FL RNA for virus assembly has been extensively documented (for review [Bernacchi, 2022]). Thus, to prevent the formation of such Gag NC-FL RNA complexes, the NC domain was deleted in the MLV-ST-MS2 construct (Fig. 2 A). The cells expressing the ΔNC showed multiple red and green dots (Fig. 2, B and C), similar to WT (Fig. 1, B and C), indicating that RNA transcription and translation were unaffected by the NC deletion. In line with what was previously published (Muriaux et al., 2004), virus release was decreased (Fig. S1) and the ΔNC virus particles did not contain FL RNA, as we showed by RT-qPCR analysis (Fig. S3). Interestingly, for the same order of red dots as in Fig. 1, B–D, red/green colocalization dropped to background level (3 ± 0.3%) after translation inhibition by puromycin (Fig. 2, B–D), indicating that colocalized red/green dots imaged with the ΔNC construct corresponded to polysomes and no longer to virus assembly events. Consequently, the MLV ΔNC was used thereafter as a reference to study MLV translation.

In conclusion, this approach allows us to detect single FL RNA polysomes (19 ± 1.2%) appearing as bright green foci, and the 24XMS2 tag enables the detection of single RNA molecule (Pichon et al., 2018). The proportion of FL RNAs under translation may seem low given the amount of structural Gag proteins required to produce viral progeny (2,000–3,000 Gag/virion [Lavado-García et al., 2021]). One cannot exclude that FL RNAs could cycle between active and inactive translation states, as already reported for other cellular mRNAs (Pichon et al., 2016). Nevertheless, the literature generally reports poor correlations between the level of mRNA and protein abundance (Greenbaum et al., 2003; Gygi et al., 1999; Tian et al., 2004). Protein abundance depends more on translation efficiency, which provides valuable information for understanding translation mechanism and protein concentration.

## Translation efficiency

Pioneering studies have demonstrated effective use of ST approach to monitor translational status of polysomes as well as translation kinetics in living cells and implemented dedicated mathematical models (Goldman et al., 2021; Morisaki et al., 2016; Pichon et al., 2016; Wang et al., 2016; Wu et al., 2016; Yan et al., 2016). To estimate the number of ribosomes in each polysome, the scFv-GFP fluorescence intensity of the translation sites is compared with that of the single, fully synthesized Gag proteins (Fig. 3 A). Assuming a uniform density of ribosomes along the mRNA, we calculated a correction factor ($X$) that considers the difference in fluorescence intensity due to different positions of ribosomes along the mRNA (Fig. S4 A):

$$\text{\# of ribosomes} = \frac{X * Median\ intensity\ of\ ST\ on\ FL\ RNA}{Median\ intensity\ of\ free\ ST\ after\ puromycin}$$
$$with\ X = \frac{N}{N - n/2}$$

where $N$ is the length of the protein counting from ST region and $n$ is the ST length.

For Gag-24ST, $X = 1.5$ (Fig. S4 A). In addition, because Gag proteins could multimerize, resulting in enhanced intensity, its fluorescence intensity was normalized to the intensity of a monomeric calibrator protein, such as the cellular KIF1C protein fused to 24ST in its C-terminal (here named: KIF1C) (Fig. 3 B and Fig. S4 B) (Pichon et al., 2021; Yan et al., 2016). Translation imaging of KIF1C was performed in our reporter system by using the same protocol and parameters. This experiment was also performed in the presence or absence of puromycin to determine median intensity of green spots corresponding to fully synthesized single KIF1C-24ST protein (Fig. 3, B and C). Then, the number of ribosomes on FL RNA was estimated as followed:

$$\text{\# of ribosomes} = \frac{X * Median\ intensity\ of\ MLV\ ST\ on\ FL\ RNA}{Median\ intensity\ of\ calibrator\ after\ puromycin}$$

Our estimation revealed that ribosome occupancy on viral FL RNA was very heterogeneous, with the majority (75%) of translating RNAs having between 1 and 4 ribosomes with a mean of 3 ribosomes (Fig. 3 D), corresponding to an average inter-ribosome distance of 1,158 nucleotides and a density of 1 ribosome per 1.1 kb. Ribosomes occupancy is governed by translation kinetics (Gilchrist and Wagner, 2006), and our results suggest a fast translation process, since ribosome density is inversely proportional to elongation rate (Huang et al., 2011).

Next, we imaged live cells to estimate ribosome elongation rate. We acquired images with a spinning disk confocal microscope for 10 min at a rate of 1 stack every 10 s, and we measured the gradual decrease of intensity occurring when a single polysome turned off (Fig. 3 E and Video 1). A mathematical model was previously implemented by Pichon et al. (2016) that assumes a constant ribosome velocity, an immediate release of nascent proteins after complete synthesis, and an undetectable lag time for scFv-GFP binding. Then, validated this estimation method by FRAP experiments (Pichon et al., 2016). We visually

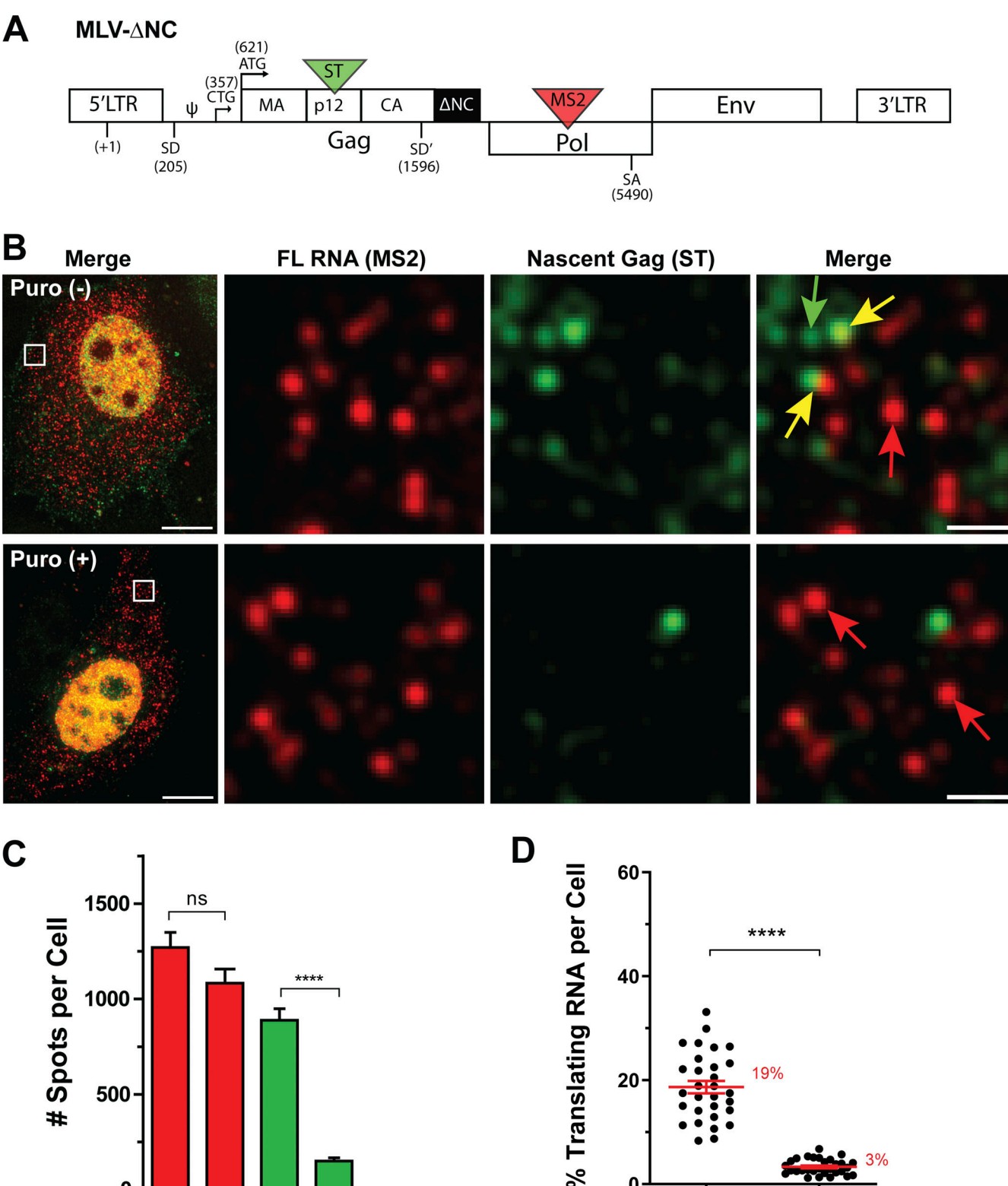

Figure 2.  **FL RNA translation of MLV-ΔNC. (A)** Schematic map of the MLV-ΔNC construct. The black box indicates the deletion of the NC domain of Gag. **(B)** Representative images of ΔNC translation without (upper panels) and with puromycin treatment (lower panels), acquired on a widefield microscope. Merge and MIP are presented. Scale bars are 10 and 1 µm for zoom insets. Arrows indicate untranslating FL RNA (red), translating FL RNA (yellow), and mature Gag (green). **(C)** Quantification of the MS2 and ST spots per cell. **(D)** Proportion of translating FL RNA. For all graphs, the mean ± SEM is shown with *n* = 30 cells. ns = nonsignificant, ****P ≤ 0.0001 (Mann–Whitney test).

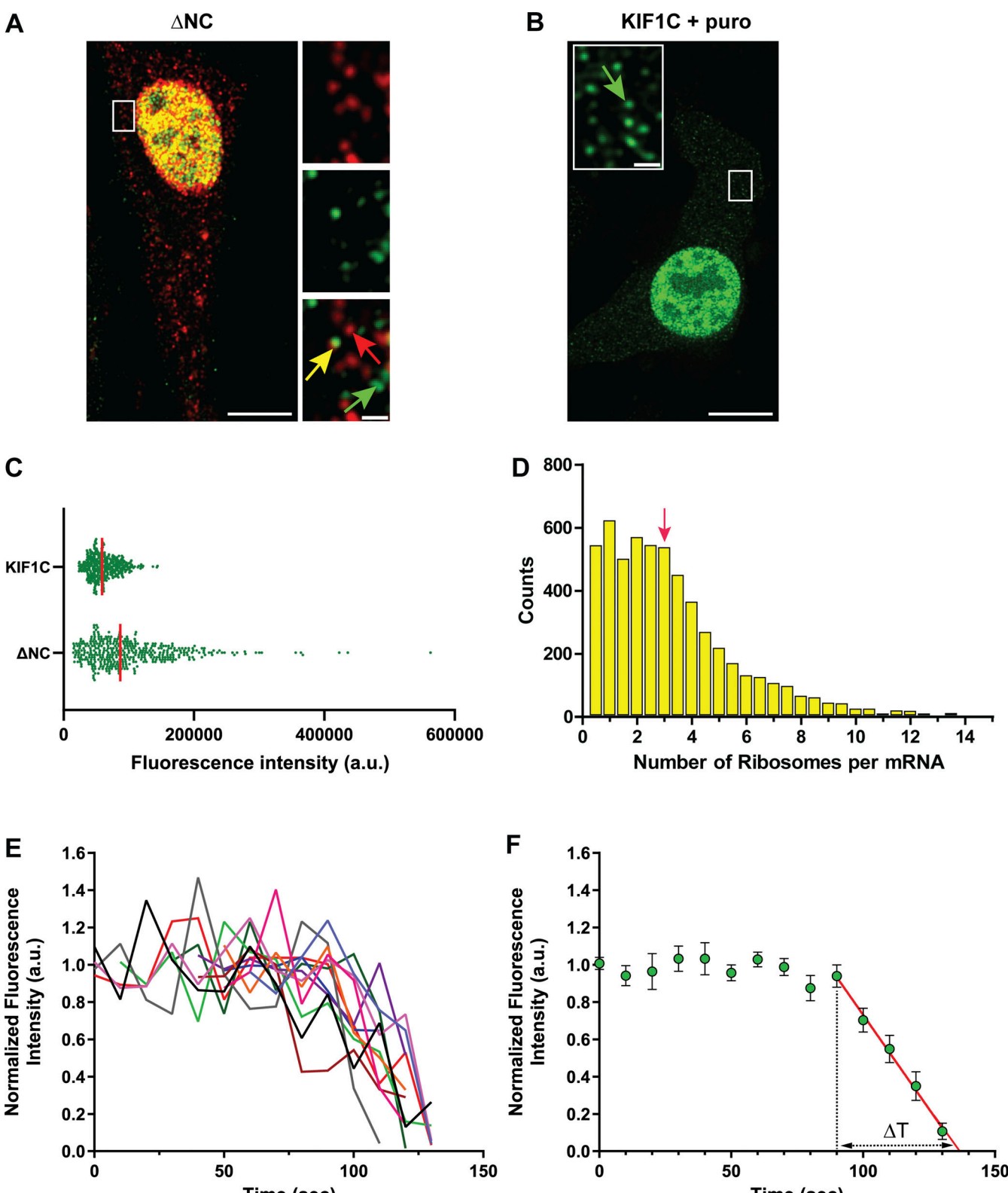

Figure 3. **Number of ribosomes per FL RNA and elongation rate. (A and B)** Representative images of a GFP/RFP cell expressing ΔNC (no puromycin) (A) and KIF1C after puromycin treatment (B) acquired on a LSM980 confocal microscope. Merge and MIP are presented. Red arrow indicates a free FL RNA molecule, yellow arrow indicates a brighter translating protein foci colocalized with the FL RNA, and green arrows indicate fully synthesized proteins. Scale bars are 10 and 1 µm for zoom insets. **(C)** Dot plot representing 400 randomly selected values of green fluorescence intensities for fully synthesized KIF1C (26,058 spots) and Gag translation sites (14,768 spots) proteins from 18 to 19 cells, respectively. **(D)** Histogram representing the frequency distribution of the number of ribosomes per FL RNA. Red arrow points the average. The number of ribosomes was determined from 5,605 spots corresponding to single polysomes, using the fluorescence intensity of calibrator, and corrected according to ribosome position (X). **(E)** Live cells analysis with the quantification of the

fluorescence intensity of individual polysomes in function of time ($n$ = 12 traces from five cells). Videos were acquired on a confocal spinning disk microscope at a rate of one stack every 10 s. Traces were normalized by the mean intensity of their plateau and aligned from the start of the fluorescence decrease. **(F)** Mean representation of traces in E ± SEM. ΔT was calculated from a linear fitting to y = 0 and starting at t + 90 s.

identified the turnoff events in the videos. The tracks were aligned to the last image just before polysome intensity decreased, i.e., the last image in which the intensity was still at a plateau. Individual curves of normalized fluorescence intensity over time of single polysomes were plotted in Fig. 3 E, and the mean curve fitted to the model was given in Fig. 3 F. The elongation rate was calculated as:

$$v = \frac{N}{t} * \left[ 1 - \frac{I(t)}{I_0} \right]$$

with $t > 0$ and $I(t = 0) = I_0$

$$v = \frac{N}{\Delta T}$$

where $I(t)$ is the fluorescence intensity over time, $N$ is the length of the protein counting from ST and $\Delta T$ is the time required for the intensity of the plateau ($I_0$) to decrease until the polysome becomes invisible. So, $\Delta T$ is deduced by extrapolating the curve to y = 0 (Fig. 3 F). Although the approach precluded rigorous quantification, the ribosome elongation rate can be estimated in the order of 22 AA/s. We also observed rare traces showing two oscillations of fluorescence intensity, meaning that the single RNA could become active again after being switched off, suggesting that single RNA could alternate between translated and untranslated states.

Taken together, our data suggest that the low level of FL RNA involved in translation (19%) could be compensated for by its fast translation kinetics, allowing the synthesis of sufficient Gag proteins to ensure viral progeny.

### Study of translation localization

Where translation takes place is a crucial question for understanding protein synthesis. Like mRNAs, which are in the right place at the right time (Buxbaum et al., 2015; Fazal et al., 2019), proteins could also be synthesized close to where they perform their function. Recent studies revealed that translation can be localized at discrete and identifiable locations (Bourke et al., 2023; Morisaki et al., 2016; Panasenko et al., 2019; Pichon et al., 2016; Shiber et al., 2018) (for review [Das et al., 2021]). A large dual protein-mRNA localization screen has shown local translation in a wide variety of subcellular compartments (Chouaib et al., 2020). The main function of the structural Gag protein is to assemble and bud at the cellular surface, and once formed, release viral particles. Electron microcopy approaches also show MLV Gag budding at intracellular membranes, leading to virus accumulation in endosomal vacuoles (Houzet et al., 2006; Sherer et al., 2003). Since, on first inspection of our ST images, FL RNAs appear evenly distributed across the cytoplasm, systematic investigations were conducted to look for possible subsets of FL RNAs (translating or not) hitchhiking on various membrane-enclosed organelles and/or tethering at the PM. To do so, we co-expressed with the FL RNA, fluorescent

turquoise proteins specific to the ER, Golgi apparatus, late endosomes (LE), lysosomes (Lyso), or PM (Fig. 4 A). A comparative study using the subcellular distribution of untranslated nonviral RNA (NV RNA) (Fig. 4 B) was used to determine the specific localization of untranslating FL RNAs (Fig. 4 C). The fraction of non-translating FL RNAs as well as control NV RNAs were determined at each location (Fig. 4 D):

Untranslating RNA association =

$$\frac{\text{\# of free red dots on turquoise surface}}{\text{\# of free red dots in whole cell}}$$

Both NV and FL RNAs were excluded from the Golgi, whereas both RNAs were detected at LE (NV = 16% and FL = 25%) and Lyso (NV = 21% and FL = 20%), suggesting weak or nonspecific localization of FL RNAs to these sites. In contrast, untranslating FL RNAs differed significantly from NV RNAs at the ER and PM. FL RNAs accumulated more at the ER (21%) than NV RNAs (9%), and the main difference between these two RNAs was at the PM, where FL RNAs were mainly localized (31%). Thus, untranslating FL RNAs specifically localized at the ER and PM (Fig. 4 D). Regarding FL RNAs undergoing translation (Fig. 4 E), their fraction was calculated for each location as follows:

Translating FL RNA association =

$$\frac{\text{\# of red and green dots colocalized on turquoise surface}}{\text{\# of colocalized red and green dots in whole cell}}$$

Translating FL RNAs showed a pattern similar to that of untranslating FL RNAs, with translation sites mainly associated with the ER (28%) and PM (26%), while a few molecules were observed at the LE (6%) and Lyso (3%) and none at the Golgi (Fig. 4 E). These results indicate that Gag synthesis takes place where the FL RNA is localized in the cell (ER and PM) and that translation does not only occur using cytoplasmic ribosomes.

### Synthesis of gGag at the ER

The mRNAs encoding gGag, which includes a peptide signal, are understood to localize and translate using ribosomes on the ER (Ahi et al., 2016). To determine whether translation sites at the ER correspond to the synthesis of gGag, two mutants were constructed to discriminate gGag from Gag synthesis. By introducing a stop codon between the $CUG_{gGag}$ and $AUG_{Gag}$ initiation codons (called MLV-ΔgGag-ΔNC), which keeps $AUG_{Gag}$ active, gGag synthesis was blocked. In this case, the colocalized signals correspond to Gag synthesis only. Conversely, by deleting the $AUG_{Gag}$, only gGag is synthesized (called MLV-gGag-ΔNC) (Fig. 5 A). Similar analyses were conducted with these two mutants by co-expressing the mTurquoise-ER protein (Fig. 5 B). The proportions of translating and non-translating FL RNAs associated with the ER were given in Fig. 5 C. When blocking the gGag synthesis (MLV-ΔgGag-ΔNC), FL RNAs were no longer associated with the ER, since untranslating FL RNA shared the same

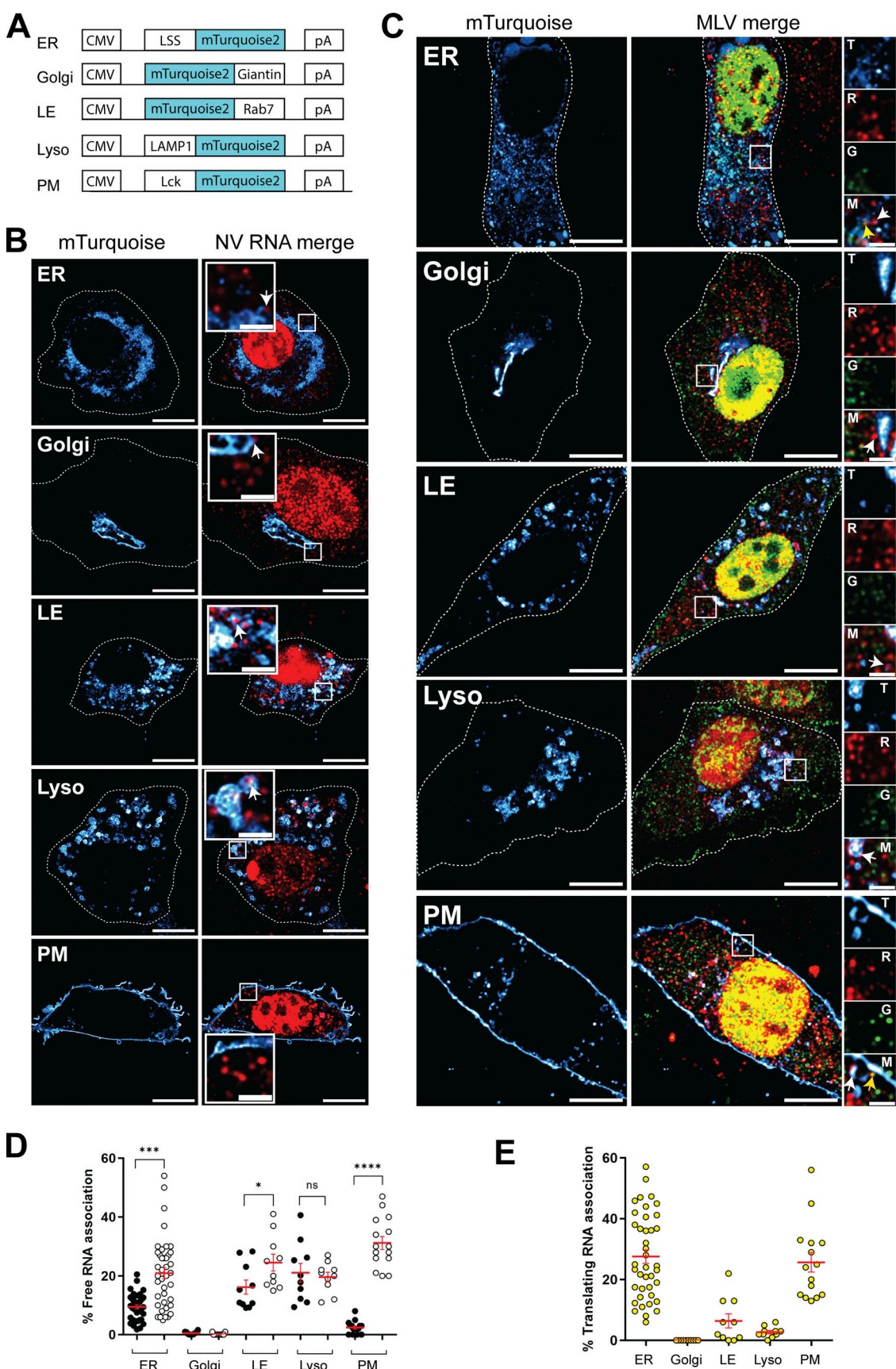

Figure 4. **Subcellular localization of non-translating (free) RNAs and translation sites. (A)** Schematic maps of the mTurquoise markers used to label different subcellular sites. **(B)** Representative images of the NIH3T3 GFP/RFP cells co-transfected with NV-MS2 and one of each mTurquoise plasmid labeling

ER, Golgi, LE, Lyso, and PM. Images were acquired on an LSM980 confocal microscope. RNA is in red and subcellular locations in hot cyan. Single z-planes are presented, and scale bars are 10 and 2 µm for zoom insets. White arrows indicate NV RNA associated with the turquoise surfaces. **(C)** Representative images of cells co-expressing ΔNC and one of each mTurquoise plasmid. Translating and non-translating FL RNAs associated with the subcellular surfaces (hot cyan) are indicated by yellow and white arrows, respectively. Single z-planes are presented, and scale bars are 10 and 2 µm for zoom insets. In zoom insets, the letters T, R, G, and M denote turquoise, red, green, and merge, respectively. **(D)** Percentage of free RNAs associated with each location determined with Imaris as the number of NV or non-translating FL RNA spots associated with each location, relative to the total number of corresponding free RNA spots in the whole cell. Black and white circles represent NV and FL untranslating RNAs, respectively. **(E)** Percentage of translating FL RNA spots associated with each location, relative to the total number of translation sites per cell. The graphs show the mean ± SEM. (n = 40 cells for ER, 15 cells for PM, and 10 cells for other locations). ns = nonsignificant, *P ≤ 0.05, ***P ≤ 0.001, and ****P ≤ 0.0001 (Mann–Whitney test).

---

level as untranslating NV RNAs. A similar drop was observed for translating FL RNA level, indicating that the ER was the site of gGag synthesis, rather than that of Gag. Consistent with these results, the non-translating FL RNA produced by MLV-gGag-ΔNC (which does not express Gag) accumulated at the ER, while the level of translation sites increased slightly compared with that of MLV-ΔNC. These results also revealed the co-translational targeting of FL RNAs, which are dedicated to gGag synthesis, to the ER.

### Local translation in the vicinity of the PM

Since the ER-associated translation sites detected by confocal microscopy fully correspond to gGag synthesis, the PM translation sites should correspond to Gag synthesis. Thus, we performed total internal reflection fluorescence microscopy (TIRFM) on fixed cells, a technique well suited to study events at the PM (Fig. S5 A). We monitored the presence of free and translating FL RNAs, using the NV RNA as negative control. Consistent with the confocal microscopy images (Fig. 4, B and D), only a few NV RNA molecules were detected in the vicinity of the PM (Fig. S5 B). Its presence is probably due to the density of signals throughout the cell. In contrast, viral free FL RNA localized there more efficiently with the detection by TIRFM of abundant red dots on the cell surface (Fig. S5 B). Next, quantification of the red and green signals revealed ∼30% of colocalized red/green dots among the total FL RNAs detected by TIRFM, confirming the presence of translation sites at or near the PM (Fig. 6, A and B). However, 10% of the colocalized signals were resistant to puromycin treatment, which could not correspond to GagΔNC–FL RNA complexes since the NC deletion abolished FL RNA recognition and consequently FL RNA packaging (Fig. S3). A possible explanation could be the difference in mesoscale fluidization between the cytoplasm and PM, while polysome collapse induced by puromycin treatment increases fluidization of the cytoplasm (Xie et al., 2024); membrane tethering or reduced mobility of PM-associated molecules could reduce the diffusivity of the fallen peptides (+puro) and thus their distance from the mRNA, resulting in a higher background of colocalized signals (10%) than that previously observed (3%) inside the cell (Fig. 2 D). Similar results were obtained with the MLV-ΔgGag-ΔNC mutant, missing gGag (Fig. 6, A and B). Experiments conducted with the MLV-gGag-ΔNC mutant revealed the absence of gGag translation signal at/near the PM (Fig. 6, A and B). In conclusion, these translation sites near or at the PM corresponded only to Gag synthesis and not to that of gGag, which is translated at the ER. Altogether, these results reveal a spatial coordination of the FL RNA and translation sites within the cell. This local translation enables the virus to control the timing and location of Gag synthesis at its site of action, the PM, which is the main assembly site of MLV.

Since a pool of FL RNA (mRNA) can be translated at the cell periphery, where another pool of FL RNA (gRNA) is encapsidated into virions, we wondered whether the proportion of translating FL RNA would change in the absence of gRNA at the PM. In a previous study, we showed that the nuclear export pathway determines the fate of FL RNA (mRNA or gRNA) and that the CRM1 pathway marks FL RNA for packaging, as leptomycin B (LMB) treatment (an inhibitor of CRM1 pathway) produces particles without gRNA (Mougel et al., 2020). Thus, we conducted experiments in the presence of LMB, which retains the gRNA pool in the nucleus (Mougel et al., 2020). TIRFM analysis of cells expressing MLV-ΔNC treated with LMB revealed no effect of LMB on the proportion of translating FL RNA at the PM (Fig. 6 C), reinforcing the dogma of the existence of two pools of FL RNA for MLV.

## Discussion

The combination of MS2-MCP and ST technologies enabled us to distinguish and quantify viral-unspliced RNA undergoing translation directly in host cell and in a viral context where the ability to produce virions was retained, albeit attenuated. Indeed, the introduction of tags into mRNA and the use of gene overexpression with transfected plasmids are the usual limitations of this approach. However, the proportion of translating RNAs did not depend on RNA spot density in cell. Viral FL RNA can engage in multiple processes, leading to translation (mRNA) and packaging (gRNA) into new viral particles, and switching between these dynamic processes must be tightly regulated for efficient viral replication. The two functions of FL RNA (mRNA and gRNA) are believed to be mutually exclusive, suggesting the existence of translation and packaging pools (Dorman and Lever, 2000). In contrast, HIV may harbor a single pool of bifunctional FL RNA (Dorman and Lever, 2000), which may explain the higher proportion of HIV-translating RNA (45%) reported by Chen et al. (2020), albeit determined in a different tissue culture system. The MLV pools were probably not the same size, as a minority of FL RNAs were under translation, despite the abundance of cytoplasmic FL RNA molecules. Untranslating FL RNAs could be maintained in a translationally repressed state due to their structure (Cap or start codon sequestration), absence of epitranscriptomic modifications (Courtney et al., 2019; Pereira-Montecinos et al., 2017), incorporation into RNP granules, or location (i.e., LE and Lyso) (Ding et al., 2021; Singh et al., 2015).

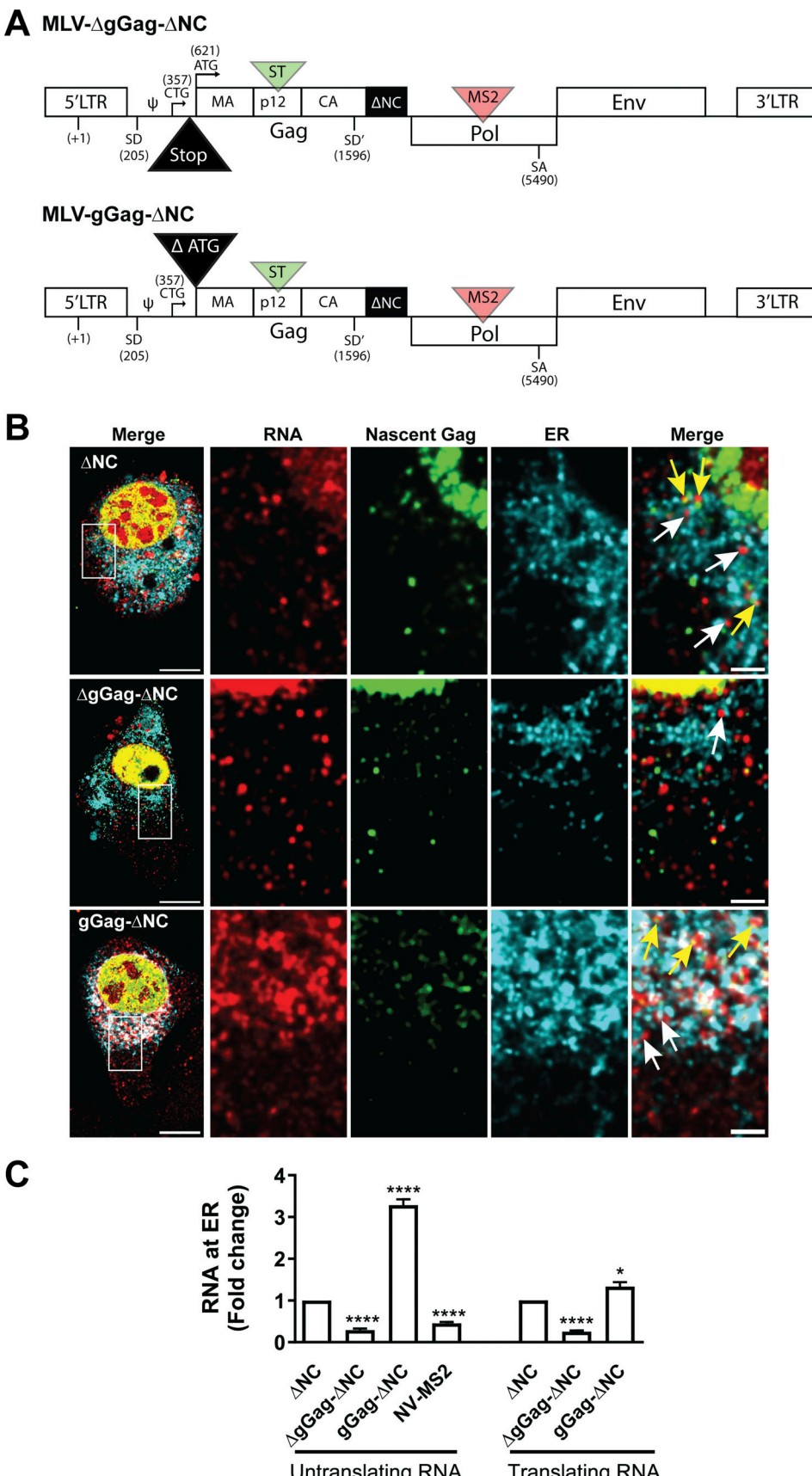

Figure 5. **Translating FL RNAs at the ER produced gGag proteins. (A)** Schematic representation of the MLV mutants. MLV-ΔgGag-ΔNC construct contains a 4-nt deletion (Δ364–367) that prevents the translation of gGag and the deletion of the NC domain. MLV-gGag-ΔNC construct lacks the AUG$_{Gag}$, preventing the

translation of Gag and deletion of the NC domain. **(B)** Representative image of cell expressing the MLV-ΔNC, MLV-ΔgGag-ΔNC, and MLV-gGag-ΔNC constructs. Images were acquired on confocal microscope. In zooms, the yellow and white arrows show the translating and non-translating RNA associated with the ER, respectively. A single z-plane is presented, and scale bars are 10 and 2 μm for insets. **(C)** Percentage of ER-associated RNA expressed as a fold change from the mean value of ΔNC. The graph shows the mean ± SEM, n = 40 cells for MLV-ΔNC and NV-MS2, and n ≥ 26 for MLV-ΔgGag-ΔNC and MLV-gGag-ΔNC. *P ≤ 0.05 and ****P ≤ 0.0001 (Mann–Whitney test).

It has also been reported that this pool of FL RNA has shorter half-life than the translation pool (Dorman and Lever, 2000). Nevertheless, mRNA and protein levels are generally poorly correlated because gene expression is regulated at multiple levels (e.g., by posttranscriptional and posttranslational modifications), and protein abundance is more dependent on the speed of the translation mechanism (Gobet and Naef, 2017; Lian et al., 2016). In mammals, ribosome elongation rate is estimated at ∼5 AA/s, using ribosome profiling (Ingolia et al., 2011) and confirmed by ST approach (Wu et al., 2016). The speed can vary depending on various factors, including the sequence and structure of the mRNA, as well as cellular conditions (viral stress responses) (reviewed in Sokabe and Fraser [2019]), making theoretical prediction of the translation rate complex (Huang et al., 2011). Therefore, it is difficult to interpret the codon usage frequency. Gag gene contains 26% of rare codons (calculated by ATGme) and 52% of GC content (42% GC in mouse genome [Mouse Genome Sequencing Consortium et al., 2002]), affecting translation rate and mRNA stabilization (Courel et al., 2019). Our experimental assessment revealed a particularly high rate of ribosome elongation along the FL RNA, which may be a way for the virus to counteract host antiviral defenses.

It is also important to consider translation in space. Although we did not specifically quantify conventional cytosolic translation, our study provides a delineation of the spatial organization of FL RNAs, translating and untranslating, on different membrane-enclosed organelles and/or tethered to the PM in single cells. Both free and translating FL RNAs were found associated with the ER and PM, suggesting that mRNA first reaches the translational localization and then, protein synthesis occurs where the mRNA is localized within the cell. In theory, the peptide signal included in gGag protein must direct the synthesis of gGag to the ER (Das et al., 2021), but this is the first time that gGag synthesis was observed directly at the ER. More surprisingly, Gag synthesis was observed at the cell periphery near or at the PM. This could explain the presence of encapsulated ribosomes in the released viral particles as previously reported (Muriaux et al., 2002). Such local translation may provide temporal control and speed of virus production (Becker and Sherer, 2017). The main advantage for the virus is that it creates a point source of high concentration of Gag protein where it is needed, avoiding the expense of expressing the protein throughout the cell when it is mainly needed at the periphery for the assembly of virus progeny. There is an additional energy benefit of transporting an mRNA compared with a protein (Bourke et al., 2023). Another benefit concerns the competition with cellular mRNA, which is weaker at the cell periphery because cellular mRNAs are mainly translated in the cytosol. However, translation

factors that modulate translation should be available at the PM. Most of them are distributed throughout the cytoplasm, but they could be translocated to the cell periphery under cellular stress, such as viral infection (reviewed in Bourke et al. [2023]). Overall, mRNA localization and local translation represent a sophisticated mechanism for the spatial and temporal regulation of protein synthesis in mammalian cells, with important implications for cell physiology, development, and disease. Although local translation is mainly observed in polarized cells (neuronal function and developmental processes), it is probably to some extent a universal feature of cells. In host–pathogen interactions, it could enable the virus to bypass the host antiviral responses.

## Materials and methods
### Cell culture and cell lines
Murine NIH3T3 fibroblast cells (CRL-1658; ATCC) and derived cell lines were maintained in DMEM supplemented with 10% FBS (Sigma-Aldrich) and 1% penicillin–streptomycin (Gibco) at 37°C and 5% CO$_2$. We confirmed cells were free of mycoplasma. The NIH3T3 cell line stably co-expressing both scFv-GFP and MCP-RFP proteins was established as follows: 1.5 × 10$^6$ cells were seeded on a 10-cm plate and transfected the following day with the plasmid pcDNA-scFv-GCN4-sfGFP-NLS (scFv-GFP), which carries a hygromycin resistance gene. After 2 days, hygromycin B (Invivogen) was added to a final concentration of 100 μg/ml. 1 wk later, transfected cells were sorted by fluorescence-activated cell sorting (Cytek Aurora CS; FACS). While maintaining hygromycin pressure, 1.5 × 10$^6$ cells were seeded on a 10-cm plate and transfected with the plasmid pcDNA-NLS-HA-MCP-TagRFP-T (referred as MCP-RFP), which carries a neomycin resistance gene. 2 days posttransfection (pt), selection for cells expressing both colors was initiated by adding G418 antibiotic (Sigma-Aldrich) to a final concentration of 600 μg/ml, in addition to hygromycin B. Following a 1-wk selection period, double-positive cells exhibiting medium levels of fluorescence for both GFP and RFP were FACS sorted. We will refer to this cell line as GFP/RFP cells hereafter.

### Plasmids and constructions
#### scFv-GFP
The plasmid pcDNA-scFv-GCN4-sfGFP-NLS (referred as scFv-GFP) expressing under the CMV promoter, the antibody against ST fused to sfGFP, GB1, and NLS, was derived from the plasmid pHR-scFv-GCN4-sfGFP-GB1-NLS (Addgene_60906 [Tanenbaum et al., 2014]). Briefly, the sequence of scFv-GCN4-sfGFP-NLS was cut with SmaI and NotI and cloned into the pcDNA3.1/hygro vector (Addgene_104355; Invitrogen) opened by EcoRV and NotI.

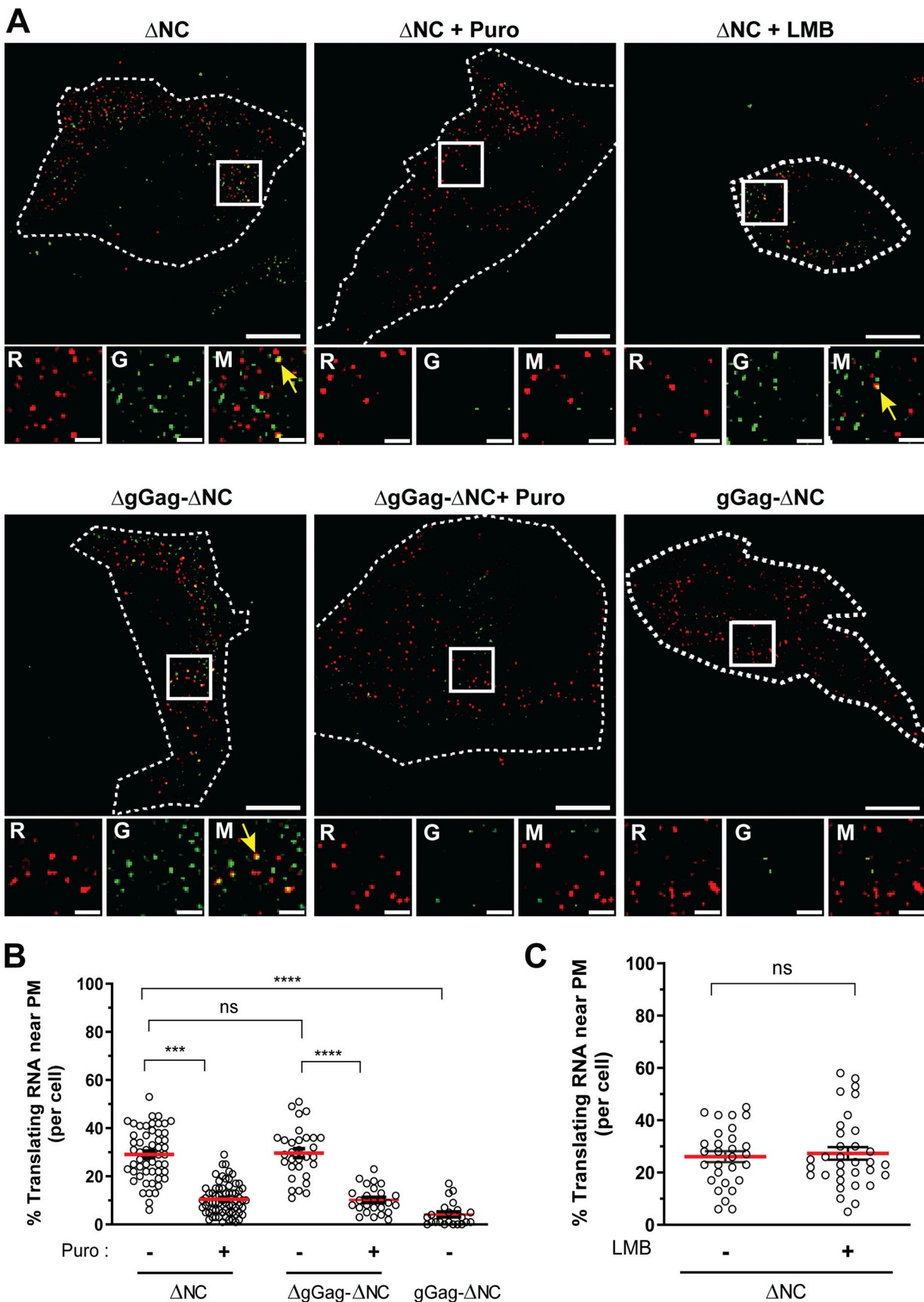

Figure 6. **Local translation of FL RNAs near the PM imaged by TIRFM. (A)** Representative images of cells expressing ΔNC, ΔgGag-ΔNC, and gGag-ΔNC with or without puromycin or LMB. Yellow arrows point to translation sites. Scale bars are 10 and 2 µm for zoom insets. The dotted line shows the boundaries of the

cell and R, G, and M indicate red, green, and merge, respectively. **(B)** Proportion of translating FL RNA near the PM. The graph shows the mean ± SEM, $n \geq 22$ cells, ns = nonsignificant, ****$P \leq 0.0001$ (Mann–Whitney test). **(C)** Effect of LMB on proportion of translating FL RNA of MLV-ΔNC. $n \geq 29$ cells.

### MCP-RFP

The plasmid pcDNA-NLS-HA-MCP-TagRFP-T (referred as MCP-RFP) encoding a nuclear version of MCP fused to RFP was derived from the plasmid pHAGE-Ubc-NLS-HA-MCP-Tag-RFP-T (Addgene_164044) described in Pichon et al. (2016). The fragment of NLS-HA-MCP-TagRFP-T was amplified by PCR with the primers forward: 5′-GCTGGCTAGCGTTTAAACTTGCGGCCG CCATGG-3′ and reverse: 5′-ACGGGCCCTCTAGATTACTTGTAC AGCTCGTCCATGC-3′ and was inserted into pcDNA3.1+/C-DYK (GenScript) opened by HindIII and XhoI by using the HiFi DNA assembly kit (NEB).

### MLV constructs

The pBSK-Eco, pMov9.1, and pRR88 plasmids correspond to the whole genome of Moloney MLV (NCBI AF033811) cloned in different vectors (Gorelick et al., 1988; Shinnick et al., 1981).

### MLV-ST-MS2

The sequence of 24XST repeats of the plasmid pcDNA4TO-mito-mCherry-24XGCN4 (Addgene_60913) was amplified by PCR (forward: 5′-CCTCACTCCTGGCGCGGCCGCGGGAGGTTCTGG AGGATTGT-3′ and reverse: 5′-GGCGCCTAGAGACGCGGCCGC GCCGCCAGACCCTCCTCG-3′) and then inserted in place of EGFP into the Gag MLV subclone pcDNA.MLVgp.EGFP.p12 (a gift from C. Baum [Voelkel et al., 2010]) opened with NotI by using the HiFi assembly kit (NEB). Next, the fragment comprising Gag-24XST was recovered using AatII and MfeI and inserted into the plasmid pBSKEco-GFP-p12-MS2 (described in Mougel et al. [2020]) previously opened with AatII and MfeI to give pBSKEco-24XST-24XMS2 (MLV-ST-MS2).

### MLV-ΔNC

The plasmid pBSKEco-24XST-ΔNC-24XMS2 (referred as MLV-ΔNC) was constructed by deleting the 168 bp of NC domain within Gag (positions T4081 to T4294). This deletion was made in a pSP72-MLV subclone using the QuikChange Lightning Site-Directed Mutagenesis Kit (210518; Agilent) using primers forward: 5′-GAGAGATGAGCAAGCTATTGACCCTAGATGACTAGG GAGGTC-3′ and reverse: 5′-GACCTCCCTAGTCATCTAGGGTCA ATAGCTTGCTCATCTCTC-3′ and then transferred to a pRR88 backbone using the restriction enzymes XhoI and ClaI to give the pRR88-ΔNC plasmid. Finally, this latter was digested with XhoI and SalI to generate a GagΔNC fragment, which was inserted into the pBSKEco-24XST-24XMS2 plasmid opened with XhoI and SalI.

### MLV-ΔgGag-ΔNC

To prevent translation of gGag, a stop codon was generated downstream the CUG of gGag by deleting 4 nt (A364-T367) in pBSKEco-24XST-ΔNC-24XMS2. Technically, the MLV-ΔNC plasmid was digested by AatII, and the 5′ overhangs were removed by Klenow fragment before religation by the T4 DNA ligase (NEB).

### MLV-gGag-ΔNC

To produce gGag, and not Gag, the translation of Gag was prevented by deleting its AUG initiation codon (AUG$_{Gag}$) in pBSKEco-24XST-ΔNC-24XMS2. Cloning was performed in two steps. First, the MLV construct p1LTR-P60-GFP (described in Akkawi et al. [2023]), which already contains the ΔAUG$_{Gag}$ deletion, was digested by Pst1, and the Pst1-Pst1 fragment was substituted into pBSKEco-24XST-ΔNC to generate the pBSKEco-ΔAUG-24XST-ΔNC, which lacks the 24XMS2 tag. Then, after XhoI digestions of the latter (containing the ΔAUG$_{Gag}$) as insert and the pBSKEco-24XST-ΔNC-24XMS2 (containing the 24XMS2 tag) as vector, the insert and vector were ligated to generate the pBSKEco-ΔAUG-24XST-ΔNC-24XMS2.

### NV-MS2

The plasmid expressing a noncoding RNA tagged with 24XMS2 repeats was previously described in Ferrer et al. (2016). Briefly, the 24XMS2 tag sequence of the RSVZ-24 plasmid (Fusco et al., 2003) was inserted into a cytomegalovirus promoter/enhancer expression vector pcDNA3.1 (Addgene_104355) to generate the NV-MS2 plasmid.

### KIF1C-ST

As previously described in Pichon et al. (2021), the KIF1C cDNA was cloned in pHRdSV40-K560-GCN4 × 24 (RRID:Addgene_72229), in place of the K560 cDNA and upstream of the 24XST, to give the KIF1C-ST plasmid.

### mTurquoise plasmids

The plasmids used to label subcellular compartments were provided by Addgene: pmTurquoise2-ER Addgene_36204), mTurquoise2-giantin (Addgene_129630), mTurquoise2-Rab7 (Addgene_112959), Lamp1-mTurquoise2 Addgene_98828), and Lck-mTurquoise2 (Addgene_98822). These plasmids were a gift from Dorus Gadella and were described in Chertkova et al. (2017), Preprint; Goedhart et al. (2012).

All plasmids containing MLV and/or tags were propagated in *Escherichia coli* stbl2 (10268019; Invitrogen) at 30°C to prevent recombination events, while the others were propagated in *E. coli* DH5α. All plasmids used in this study have been fully sequenced using Nanopore sequencing (Plasmidsaurus).

### DNA transfection and drug treatments

To determine the proportion of translating RNA, GFP/RFP cells (250,000 cells per well) were grown on glass 0.5% gelatin-coated coverslip (square 20 mm 1.5) (Marienfeld) in 6-well plate and transfected with 2 μg of MLV-ST-MS2 or control pSP72 (empty vector, GenBank X65332; Promega) plasmid at equimolar ratios, using JetPRIME or JetOPTIMUS (Polyplus) reagent, following the manufacturer's recommendations. To study RNA localization, 100 ng of plasmid expressing a turquoise marker specific to a subcellular compartment was co-transfected with the MLV-ST-MS2 or NV-MS2 control plasmid

at a molar ratio of 1:20. 4 h pt, cells were washed with warm PBS and grown in fresh medium. At ~48 h pt, the medium was removed, and the cells were washed with warm PBS. To inhibit translation, fresh medium containing Puromycin (Invivogen) at a final concentration of 100 μg/ml or cycloheximide (Sigma-Aldrich) at a final concentration of 200 μg/ml for 15 min at 37°C was added to the cells. Inhibition of nuclear RNA export by CRM1 was performed by treating cells with 5 nM of LMB (LC Laboratories) for 20 h as described in Mougel et al. (2020). Subsequently, cells were fixed with 4% formaldehyde (Sigma-Aldrich) in PBS and incubated for 10 min at room temperature and then washed twice with PBS. The coverslips were carefully recovered and allowed to dry before being mounted on slides (StarFrost) using Vectashield (Vector Laboratories) with or without DAPI (Thermo Fisher Scientific) (dilution 1:2,000) and then sealed using nail polish. Finally, the slides were stored in a light-protected environment at 4°C.

## Quantification of FL RNA by RT-qPCR

NIH3T3 cells ($1.5 \times 10^6$) were seeded on a 10-cm plate, and the day after, they were transfected with 10 μg of MLV molecular clone pRR88-WT or pRR88-ΔNC using JetPEI reagent. RNA was extracted from transfected cells with TriReagent (MRC) according to the manufacturer's instructions. Virions were purified from 10 ml of filtered culture supernatants by centrifugation through a 20% sucrose cushion at 30,000 rpm for 1 h 30 at 4°C in an SW32 rotor. Pellets were resuspended in DMEM with 8 U of DNase RQ1 (Promega) and incubated at 37°C for 45 min to reduce contamination by the transfecting plasmid DNA. Then, TES 4X (200 mM Tris, pH 7.5, 20 mM EDTA, 0.4% SDS) and 20 μg of tRNA carrier were added to the virions before extraction of the nucleic acids by phenol/chloroform and ethanol precipitation. RNA samples from cells and virus were treated with RQ1 DNase in the presence of RNasin (Promega) for 25 min at 37°C to remove DNA contamination, followed by phenol–chloroform and chloroform steps, and finally RNAs were precipitated with ethanol. After 70% ethanol wash, RNA pellets were dissolved in RNAse free water, and RNA was quantitated by measuring optical absorption at 260 nm (as described in Chamontin et al. [2012]).

For RT-qPCR, the RT step was performed with Superscript III reverse transcriptase (Thermo Fisher Scientific) and initiated with an oligo (dT) primer. A control RT experiment was systematically performed without enzyme to check the absence of DNA contamination. Quantitative PCR assay was achieved with 2.5% of RT reaction, the SYBR Green kit (Roche), and the following primers: sMLV3350: 5′-TATCGGGCCTCGGCAAGAAAG-3′ sense and aMLV3600: 5′-AAACAGAGTCCCCGTTTTGGTG-3′ antisense for FL RNA and sGAPDH721 5′-GCTCACTGGCATGGC CTTCCGTGT-3′ sense and aGAPD921 5′-TGGAAGAGTGGGAGT TGCTGTTGA-3′ antisense for GAPDH mRNA, which was used to normalize FL RNA extracted from cells. The products were amplified by 35 cycles: 95°C for 15 s; 60°C for 15 s, and 72°C for 20 s using the RotorGene (Labgene) systems as described in Chamontin et al. (2012). The measured RNA copy numbers are corrected with the mock (copy numbers determined from cells transfected with the empty Sp72 plasmid) and scaled to the total

number of cells present in the dish and to the total virions present in 10 ml of cell supernatant.

## Western blotting of virions

$1.5 \times 10^6$ NIH3T3 cells were seeded on a 10-cm plate, and the day after, they were transfected with 20 μg of MLV-ST-MS2 plasmid, using JetPRIME reagent. At 48 h pt, the culture medium was collected for virus analysis. The medium was centrifuged at $1,500 \times g$ for 5 min and filtered through a 0.45-μm pore-size filter to eliminate cell debris before ultracentrifugation on a 20% sucrose/PBS cushion at $33,000 \times g$ for 1.5 h at 4°C. Virus-containing pellets were resuspended in 50 μl of sample buffer (80 mM DTT, 0.5% bromophenol blue, 8% SDS, 40% glycerol, 250 mM Tris HCl [pH 6.8], and 20% β-mercaptoethanol) and loaded on 8% SDS-PAGE. Proteins were electro-transferred onto PVDF membrane. Gag was detected with a rat anti-capsid antibody (1/500, hybridoma H187, from B. Chesebro) (Chesebro et al., 1983), a peroxidase-conjugated (HRP) secondary antibody, and goat anti-rat IgG (1/2,000; Invitrogen) (Cat #NB 7115, Novus). After incubation with ECL Clarity Max substrate (Bio-Rad), the fluorescence was recorded by a CCD chemiluminescence camera system (ChemiDoc MP Imaging System; Bio-Rad).

## Image acquisition of fixed cells

To determine the proportion of translating RNA in cells, fixed cells were imaged on a Zeiss AxioImager Z2 widefield microscope equipped with a scMOS ZYLA 4.2 MP camera, using a 100× oil objective (Plan Apochromat; 1.4 NA) and controlled with MetaMorph (RRID:SCR_002368). DAPI, GFP, and DsRed filters were used, and excitation time was set to 20, 300 and 400 ms, respectively. Area of 600 × 600 pixels were acquired, with each pixel corresponding to a size of 65 nm and z-step of 0.3 μm with a minimum of 15 planes. To study translation efficiency and localization, fixed cells were imaged using a laser scanning confocal microscopy (LSM980 NLO; Zeiss) controlled by Zen 3.5 and equipped with an Objective Plan-Apochromat 63×/1.40 oil DIC M27. Scanning was performed at a resolution of 1,024 × 1,024 pixels and a magnification of 2× yielding a pixel size of 66 nm. Laser lines, detection wavelengths, and detectors were set as follows: laser 405 nm at 1% power and detection in 440–520 nm using Multialkali-PMT detector, laser 488 nm at 1% power and detection in 490–570 nm using GaAsP-PMT detector, and laser 561 nm at 8% detection in 561–698 using Multialkali-PMT. Detector gain was set to 1, scan direction as bidirectional, pixel time of 2.05 μs, and z-step of 0.2 μm (20 planes minimum).

TIRFM was performed on an inverted Eclipse Ti microscope (Nikon), equipped with a 100×/1.49 NA TIRF APO oil immersion objective and an EMCCD iXon 897 camera (Ultra Andor). Transfected cells were seeded on 25-mm glass coverslips coated with 0.5% gelatin. The coverslips were transferred onto an At-tofluor Cell Chamber (Invitrogen) containing 1 ml of PBS. Using the iLas 2 TIRF illuminator module (GATACA Systems) in MetaMorph, illumination was switched to TIRF mode to excite selectively molecules within ~100 nm of the coverslip. Tetra-Speck fluorescent microspheres, 0.1 μm (Thermo Fisher Scientific) deposited on a coverslip were used to verify the TIRF mode

(Fig. S5). Imaging of GFP and RFP was achieved by exciting with 488- and 560-nm lasers (300 and 400 ms of exposure at 100 mW of power and detection windows set to 500–550 and 570–620 nm, respectively) with an acquired area of 512 × 512 pixels and a pixel size of 160 nm.

### Live-cell imaging

Cells transfected with MLV-ST-MS2 plasmid were plated (30,000 cells per well) onto a μ-Slide 8-well glass bottom #1.5H (ibidi), coated with collagen (30%) (Sigma-Aldrich), and imaged at 37°C with $CO_2$ using a Dragonfly spinning disk confocal microscope (Nikon) controlled by Fusion software and equipped with a 100×/1.45 NA DT 0.13-mm oil objective and an EMCCD iXon888 camera (Andor). Lasers 488 and 561 were set to 30% of power, and the acquisition was set to 50 ms exposure at a rate of 60 stacks every 10 s, with z-step of 0.6 μm (10 planes minimum per stack). Two hours before imaging, DMEM was replaced with FluoroBrite DMEM (Thermo Fisher Scientific) supplemented with 10% FBS.

### Image analysis
#### Deconvolution

Before deconvolution, videos were corrected for photobleaching in Huygens Essential version 22.10 using automatic parameters. All confocal images and videos were deconvolved with Huygens. The deconvolution was done using the CMLE algorithm with automatic parameters: background subtraction, acuity, and signal-to-noise ratio, with a maximum of 26 iterations and a threshold of 0.01.

#### Spots detection and colocalization

Images were analyzed using IMARIS Bitplane software version 10.0. Image stacks were projected using the 3D view mode, and the spot detection tool was used to identify and quantify the red (RNA) and green (ST) spots in the cytoplasm of every cell. Spot creation wizard was used with the following parameters for ST spots: XY diameter = 0.4 μm, Z size = 0.6 μm and for RNA spots: XY diameter = 0.3 μm and Z size = 0.6 μm. Nonspecific detected spots were filtered using the quality filter and manually thresholded using the viewer tool. Spots outside the cell and inside the nucleus were excluded. RNA/ST colocalization were quantified to determine the proportion of translating RNA. Spots were defined as colocalized if the shortest distance to the nearest neighbor was ≤0.52 μm (as described in Chen et al. [2020]) using the colocalization tool in Imaris. The total fluorescence intensity of the ST spots was recovered to calculate the number of ribosomes. To analyze the videos, colocated spots were manually added using a spot diameter of 0.6 μm (XY) and a Z size of 0.6 μm, and then the total fluorescence intensity as a function of time was recovered to calculate the elongation rate. TIRFM images were analyzed as follows: a Laplacian of Gaussian filter was applied to the TIRF images using Fiji, and then colocalization analysis was performed in Imaris with spot diameter (XY) set to 0.5 μm for green and red channels. Cell boundaries were manually added by drawing a line and then using the dotted line plugin in Fiji.

### Surface rendering and RNA localization

All subcellular compartments were delimited using the surfaces creation wizard in Imaris using the fluorescence of DAPI and mTurquoise2 channels, with surface detail set at 0.5 μm for the nucleus and 132 nm for the other compartments. RNA spots were separated into two groups: translating RNA (RNA colocated with ST) and non-translating RNA (free RNA signal). Each of these groups were colocated with the surface previously created of the subcellular compartments, and they were considered colocated when the distance between the RNA spot and the surface was = 0 μm.

### Linear unmixing

To overcome small fluorescence bleed through of mTurquoise2 fluorophore with the 488-nm laser into the GFP channel, we performed linear unmixing (Zimmermann et al., 2003) after image acquisition using Zeiss software version 3.5. For this purpose, NIH3T3 cells were transfected with each plasmid-expressing mTurquoise fluorophore or the scFv-sfGFP-NLS plasmid. Cells expressing Turquoise or GFP were imaged on a LSM980 confocal microscope multi-PMT spectral detector using the spectral mode with a chosen bandwidth of 8.9 nm to determine their experimental emission spectrum, using the 405-nm (1% of laser power) and 488-nm (1% of laser power) laser lines for mTurquoise and GFP, respectively, with a detection window ranging from 415 to 663 nm and 415 to 601 nm for mTurquoise and GFP, respectively. Experimental emission spectra were saved to a database and then used to perform linear unmixing, using the linear unmixing tab in Zeiss 3.5 on the images where mTurquoise and sfGFP were acquired. This linear unmixing process effectively eliminated the unwanted bleed through, allowing us to accurately quantify and analyze the fluorescence signals from the GFP and mTurquoise2 channels.

### Statistical analysis

The statistical significance of data was evaluated using a non-parametric $t$ test (Mann–Whitney) in GraphPad Prism v10. P values are presented as follows: *$P \leq 0.05$, **$P \leq 0.01$, ***$P \leq 0.001$, and ****$P \leq 0.0001$ and are indicated in figure legends. All data are expressed as mean ± SEM.

### Online supplemental material

Fig. S1 shows the virions released from cells expressing the tagged MLV molecular clones used in the study by using western blot assay with anti-Gag antibody. Fig. S2 explains the methodology with the establishment of stable cell lines (A and B) and the microscopic analysis and image processing (C). There are controls for quantification (D) and specificity (E) of the signals. Fig. S3 shows that virions released from cells expressing ΔNC MLV did not contain FL RNA, as monitored by RT-qPCR. Fig. S4 illustrates the variation of fluorescence intensity during ribosome progress on FL RNA depending on the ST position within the gene (A). The Kif1C-ST contains 24xST at the C terminus, and it is used as a control for fluorescence emitted as monomeric protein (B). Fig. S5 shows the difference between epifluorescence and TIRF microscopy (A). Using TIRFM, ΔNC FL RNA was abundant at the cell periphery, whereas no control RNA was

observed (B). Video 1 shows the live imaging of FL RNA translation with a special focus on polysome turned off.

## Data availability
Data and materials are available from the corresponding author upon reasonable request. All unique reagents generated in this study are available from the corresponding author with a completed Materials Transfer Agreement.

## Acknowledgments

We thank the Montpellier RIO Imaging platform for assistance with microscopy and FACS. We thank V. Baecker for assistance with image analysis. We are also grateful to J. Feuillard for technical assistance. We thank Dr Y. Bare and R. Mensingh for critical reading of the manuscript.

This work was partially supported by the Centre National de la Recherche Scientifique, the University of Montpellier, and the French National Research Agency. F. Leon-Diaz received fellowships from the Infectiopole Sud and the French National Agency for Research on AIDS and viral hepatitis.

Author contributions: F. Leon-Diaz: conceptualization, data curation, formal analysis, investigation, methodology, visualization, and writing—original draft, review, and editing. C. Chamontin: formal analysis, investigation, validation, visualization, and writing—original draft, review, and editing. S. Lainé: conceptualization, data curation, investigation, validation, visualization, and writing—review and editing. M. Socol: formal analysis and validation. E. Bertrand: conceptualization and supervision. M. Mougel: conceptualization, data curation, funding acquisition, investigation, methodology, project administration, resources, supervision, validation, and writing—original draft, review, and editing.

Disclosures: The authors declare no competing interests exist.

Submitted: 15 May 2024

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

# Supplemental material

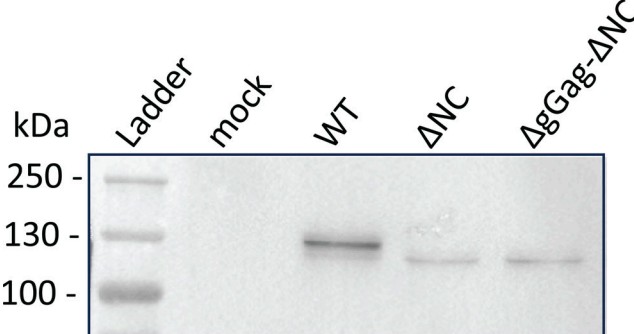

Figure S1.   **Analysis of virions released by the GFP/RFP cell line–expressing tagged MLV constructs by using western blotting.** The Gag-24ST protein was detected using the anti-CA antibody at the expected size (WT: 134 kDa and mutants: 127 kDa). Source data are available for this figure: SourceData FS1.

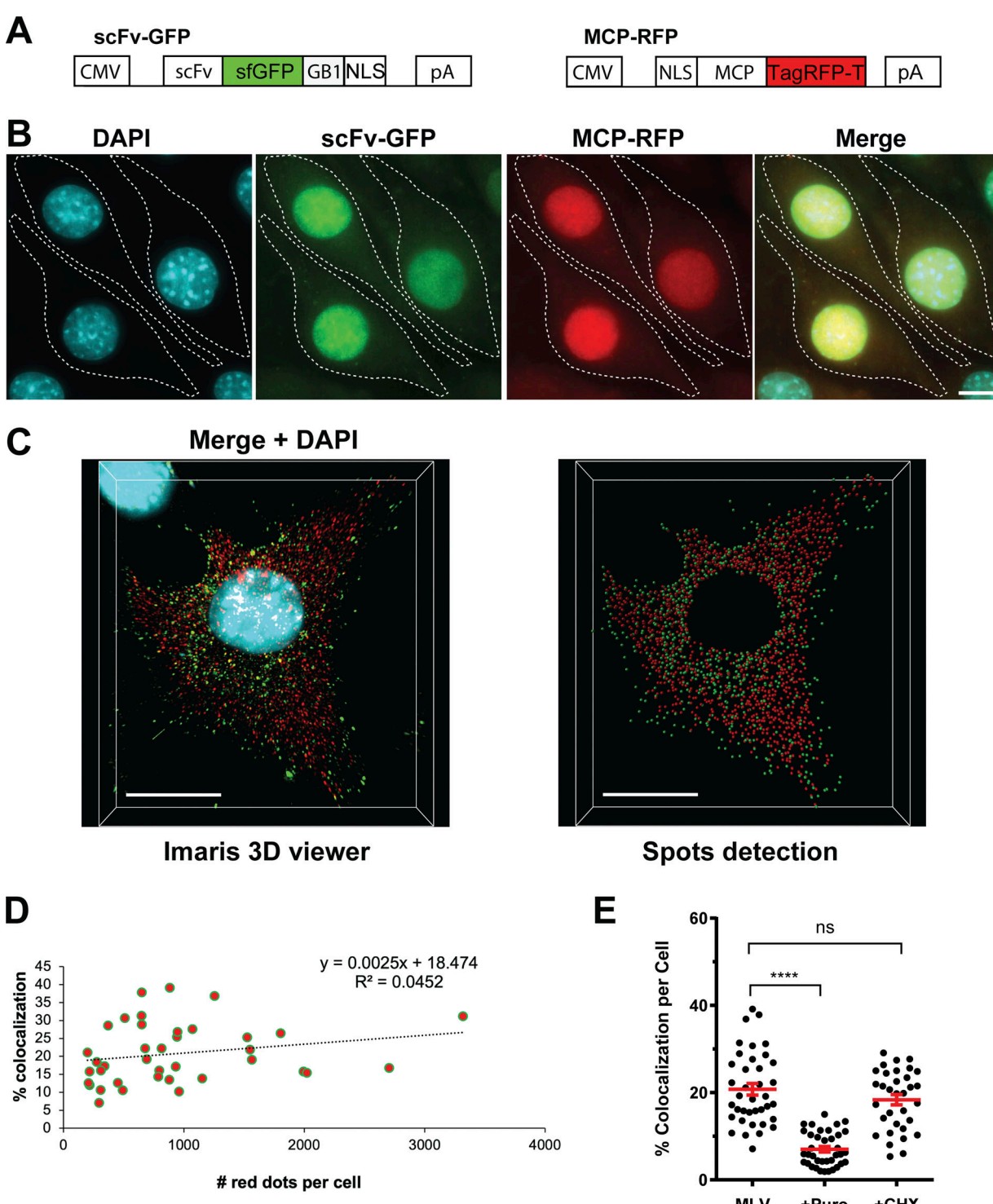

Figure S2. **Translation monitoring and controls. (A)** Methodology for labeling and microscopy schematic maps of the two plasmids, scFv-GFP and MCP-RFP, encoding fluorescent proteins that bind the ST and MS2 tags, respectively, both fused to an NLS. **(B)** Representative images of the GFP/RFP cell line, expressing the scFv-GFP (green) and MCP-RFP (red) proteins, acquired on a widefield microscope. DAPI staining is in cyan, and scale bar is 10 μm. **(C)** Spot detection with Imaris for nascent Gag (green) and RNA (red). Stack images of a GFP/RFP cell expressing MLV-WT are projected in 3D using the 3D viewer tool in Imaris (on the left). DAPI staining is in light blue. The image on the right shows the spots detected for nascent Gag (green) and RNA (red). Spots outside of the cell were manually removed and spots in the nucleus were excluded using a surface created with DAPI channel. Scale bars are 10 μm. **(D)** Analysis of the dependence of the red/green spots colocalization on the total number of RNA molecules in the cell. The slope (0.0025) of the simple linear regression function and the low value of the coefficient of determination ($R^2$ = 0.04) indicate a very weak dependence of the percentage of colocalization on the abundance of the red dots in the cells ($n$ = 38 cells). **(E)** Inhibition of WT FL RNA translation. Comparison between the effects of puromycin and cycloheximide (CHX), which have distinct mechanisms of action. Colocalized red–green dots were quantified, and the proportion of translating FL RNA per cell was calculated for $n \geq 33$ cells. The graph shows the mean ± SEM, ns = nonsignificant and ****$P \leq 0.0001$ (Mann–Whitney test).

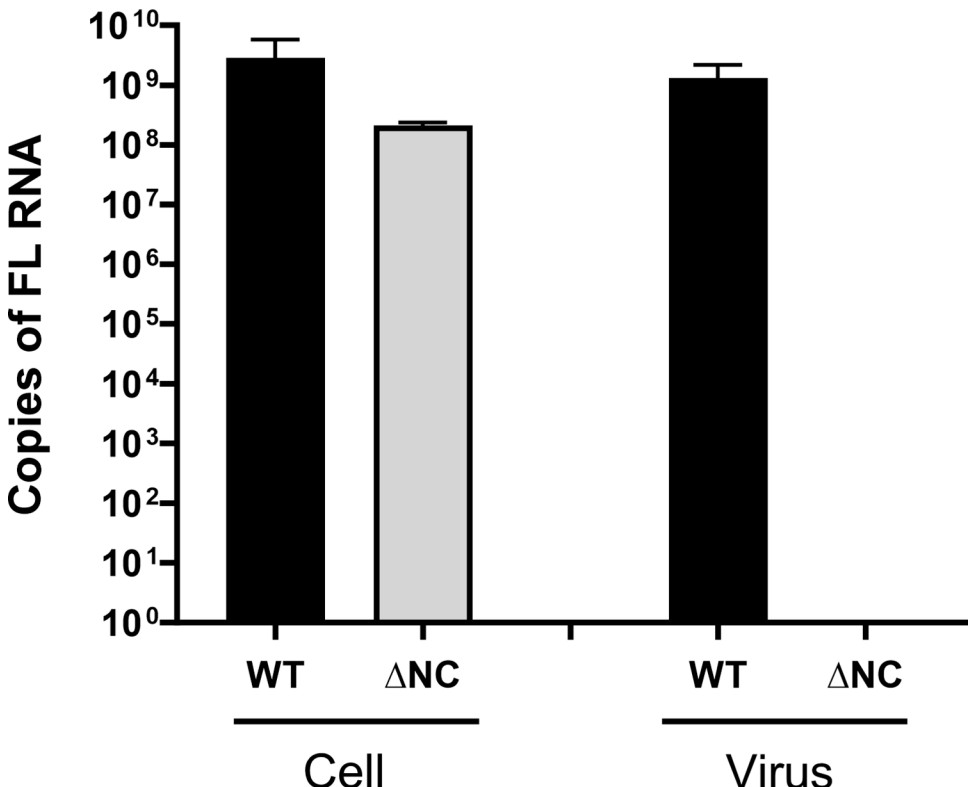

Figure S3. **Effect of ∆NC mutation on FL RNA packaging into viruses.** FL RNA was quantitated by RT-qPCR in NIH3T3 cells transfected with the MLV molecular clone: pRR88-WT or pRR88-∆NC (described in Grigorov et al. [2007]) or with an empty pSP72 vector (as control) and in the released viral particles. Mock controls were subtracted from assays, and the measured RNA copy numbers were scaled to the total number of cells present in the dish and to the whole-cell supernatant. The graph shows the mean ± SD, $n$ = 2 independent transfections.

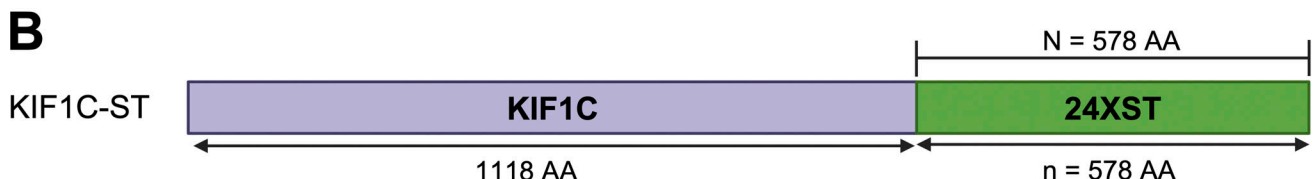

Figure S4. **Parameters used to determine the number of ribosomes translating MLV FL RNA. (A)** Illustration of different intensities of fluorescence of a single ribosome depending on its position on FL RNA. Lengths of Gag-24ST domains are given in amino acids. Created with https://BioRender.com. **(B)** KIF1C calibrator fused to 24XST at C terminus. *N* is the length of the protein counting from ST region and *n* corresponds to the length of 24XST in amino acids.

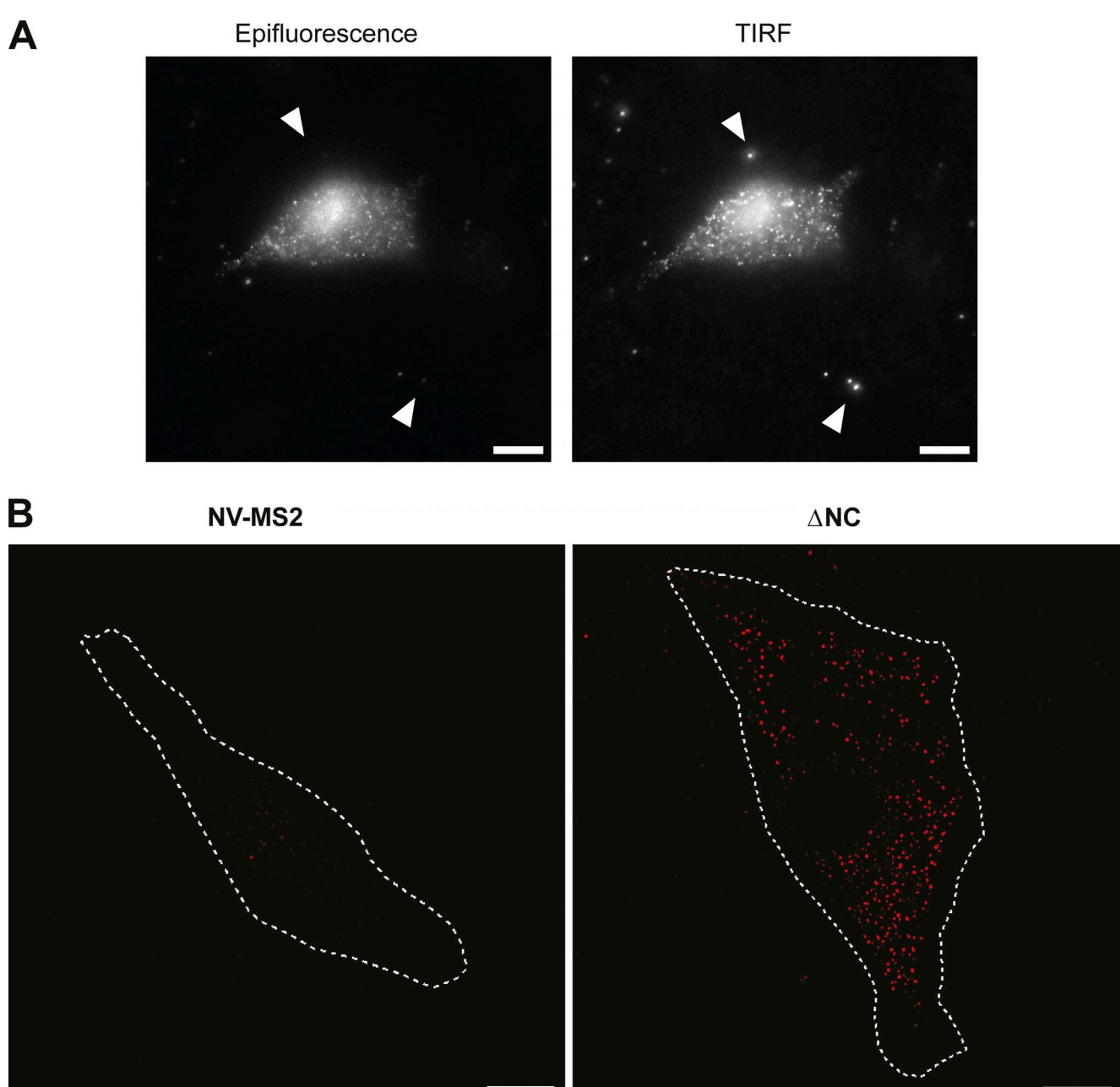

Figure S5.   **RNA imaging imaged by TIRFM. (A)** Control for TIRF mode with fluorescent beads. NIH3T3 GFP/RFP cells were transfected with ΔNC and fixed. Red channel is shown in gray. White arrows indicate the fluorescent beads (100 nm) deposited on the coverslip a few minutes before acquisition. Epifluorescence (left) and TIRF (right) were captured at the same focal plane. Raw images are presented, and scale bar is 10 μm. **(B)** Representative images of FL and NV RNAs at the periphery of GFP/RFP cells expressing MLV-ΔNC or NV-MS2. Scale bars are 10 μm, and the dotted line indicates the boundaries of the cell.

Video 1.   **Live imaging of FL RNA translation with a special focus on polysome turned off.** NIH3T3 GFP/RFP cell expressing MLV-ΔNC was imaged on a spinning disk confocal microscope (1 stack per 10 s, 60 stacks in total, with a z space of 0.6 μm). The video is a zoom with red and green channels presented separately and corresponding to RNA and nascent Gag, respectively. The circle emphasizes the quantified ST spot, which is presented as black curve in Fig. 3 E. MIP projection is presented, and the video is accelerated 10 times.

