## [Peer Review File · The Journal of Cell Biology]

Translation of unspliced retroviral genomic RNA in the host cell is regulated in both space and time

Felipe Leon Diaz, Celia Chamontin, Sebastien Lainé, Marius Socol, Edouard Bertrand, and Marylene Mougel

Corresponding Author(s): Marylene Mougel, French National Centre for Scientific Research

Review Timeline:

Submission Date:	2024-05-15
Editorial Decision:	2024-07-12
Revision Received:	2024-11-06
Editorial Decision:	2024-12-14
Revision Received:	2024-12-20

Monitoring Editor: Jens Lykke-Andersen

Scientific Editor: Dan Simon

Transaction Report:

DOI: <https://doi.org/10.1083/jcb.202405075>

July 12, 2024

Re: JCB manuscript #202405075

Dr. Marylene Mougel
French National Centre for Scientific Research
IRIM-UMR9004
1919 Route de Mende
Montpellier, France 34293
France

Dear Dr. Mougel,

Thank you for submitting your manuscript entitled "How and where is the unspliced retroviral genomic RNA translated in the host cell." The manuscript has been evaluated by expert reviewers, whose reports are appended below. Unfortunately, after an assessment of the reviewer feedback, our editorial decision is against publication in JCB.

You will see that although the reviewers express divergent opinions on the suitability of the study for JCB they all raise important concerns that would require a significant amount of new experiments to address. The most important question is whether the observed gRNA translation at the plasma membrane contributes to viral particle assembly. In addition, it would be necessary to address the comments on additional, orthogonal validation (beyond puromycin controls) that your assays are in fact monitoring translating gRNAs as opposed to Gag-associated gRNAs (Reviewer #1, Figs. 1 & 2 comments, Reviewer #2, comment #2), and of course the other minor comments on clarity of the text and figure legends.

Although your manuscript is intriguing, we feel that the points raised by the reviewers are more substantial than can be addressed in a typical revision period. If you wish to expedite publication of the current data, it may be best to pursue publication at another journal.

Given interest in the topic, we would be open to resubmission to JCB of a significantly revised and extended manuscript that fully addresses the reviewers' concerns and is subject to further peer-review. If you would like to resubmit this work to JCB, please contact the journal office to discuss an appeal of this decision or you may submit an appeal directly through our manuscript submission system. Please note that priority and novelty would be reassessed at resubmission of the revised manuscript.

Regardless of how you choose to proceed, we hope that the comments below will prove constructive as your work progresses. We would be happy to discuss the reviewer comments further once you've had a chance to consider the points raised in this letter. You can contact the journal office with any questions at cellbio@rockefeller.edu.

Thank you for thinking of JCB as an appropriate place to publish your work.

Sincerely,

Jens Lykke-Andersen, PhD
Monitoring Editor
Journal of Cell Biology

Dan Simon, PhD
Scientific Editor
Journal of Cell Biology

Reviewer #1 (Comments to the Authors (Required)):

The manuscript entitled "How and where is the unspliced retroviral genomic RNA translated in the host cell" by Leon-Diaz and coworkers explores the fate of MLV genomic RNA, focusing on its translation and cellular localization using single-molecule microscopy. They developed MLV molecular clones that enable simultaneous detection of RNA (via MS2 tag) and its translation (via SunTag), a technique previously applied to HIV-1. Their findings reveal that only 19% of viral RNA serves as mRNA, with an average of three ribosomes per RNA molecule. This low ribosome density is offset by a fast translation rate (22 amino acids per second). Interestingly, viral RNA translation occurs not only in the cytoplasm but also in locations such as the endoplasmic reticulum (ER) and plasma membranes (PM). By inhibiting the translation of the glycoGag (gGag) protein from an upstream

CUG codon, they demonstrated that gGag is preferentially translated at the ER, while Gag translation occurs at or near the PM. Overall, these results indicate that the intracellular location of viral RNA is linked to its translation site.

This study significantly contributes to our understanding of retroviral RNA fate by redefining the translation locations. It will likely interest a broad range of researchers. I have a few suggestions and comments to strengthen the manuscript.

Major comments.

- Figure 1: Did the authors try using another translation inhibitor, such as cycloheximide? Cycloheximide stalls ribosomes on mRNA by binding to the ribosome E site and interfering with polypeptide elongation, so the proportion of dual-colored RNA should not decrease with this inhibitor.
- Figure 2: The authors suggest that the MLV Δ NC construct can be used as a reference to study MLV translation. While I mostly agree, could they measure the amount of gRNA present in viral particles, given that figure S1 shows MLV Δ NC can still form virion progeny (empty or full)? This would mean that Gag Δ NC could still interact with viral RNA.
- Figure 3: What is the translation rate on MLV WT RNA? Is it different from MLV Δ NC? Could the NC domain influence this rate?
- Figure 5: Did the authors build a mutant where the main Gag AUG initiation codon is substituted to inhibit its translation and favor gGag translation? In this case, do they only observe translation at the ER?
- The authors concluded that MLV translation can occur at the PM or cell periphery but they did not discuss the coexistence of two pools of RNA (for translation and packaging) in MLV. Do they think PM-translating MLV mRNAs are the ones to be encapsidated? This would challenge the current understanding of MLV RNA pools and provide a compelling conclusion to the study.

Minor comments:

- Figure 4C: please define the color dots (white and black).
- Figure 5A: the name of the vector is not correct, it should indicate: MLV- Δ gGag- Δ NC. Please check as well in the corresponding text
- Figure 6A: it is indicated in the legend that left panels are in absence of puromycin but the lower left panel is indicated "Puro (+)".
- The experimental descriptions do not always clearly indicate whether puromycin was added in each experiment.

Reviewer #2 (Comments to the Authors (Required)):

In this manuscript, Leon-Diaz et al. examined MLV unspliced RNA translation using the established SUNTAG and MS2 approach to visualize translated protein and viral RNA. This strategy has been used to study translation of multiple RNAs. The authors concluded that only 20% of the RNA is being translated and translation is coordinated spatially near the plasma membrane to facilitate virus assembly. In general, this manuscript suffers from not validating results with other approaches and over-interpreting the results. Although complex imaging approaches were used, this study adds little to our current understanding of translation or MLV biology.

Major issues:

1. The authors claim that less than 20% of the unspliced RNA is being translated. This is likely to be a significant underestimation because of their experimental design, which only allows visualization of a small portion of the gag gene. The authors could validate their measurement using a standard method to measure the percentage of MLV RNA in polysomes. Without such validation, their conclusion is unsupported.
2. The authors examined MLV RNA location in five compartments and concluded that RNA is translated in the ER and near the plasma membrane but not in the Golgi, lysosome, or late endosomes. From this, they conclude "translation is regulated in both space and time." Whole-cell imaging showed that MLV RNA is present throughout the cell and there is a large cell-to-cell variation in translation rate (up to 8-fold according to Fig. 1 and 2). To understand whether translation is regulated in space, the comparison should be between RNAs in one of these compartments versus elsewhere in the same cell. The comparison as it currently stands does not tell the whole story.
3. The significant presence of colocalized RNA and Gag signals (one-third of total detected signals) after puromycin treatment (Fig. 6) suggests these are not translating RNA but Gag:RNA complexes. This very high background is unexpected and should be investigated.

Reviewer #3 (Comments to the Authors (Required)):

In this manuscript Felipe Leon-Diaz et al have investigated the translational activity of viral RNA by using dual labelling system of MS2-MCP and ST technologies. These technologies have allowed them to visualize single molecules of RNA undergoing translation in infected cells. Using MLV system, they were able to visualize site of translation of the viral RNA when it is prevented from interacting with NC and hence prevented from being actively engaged in assembly. They have made several interesting observations, including the fact that the site of translation of mRNA includes places near Plasma membrane where assembly is taking place and synthesis of glycoGag at the ER.

The main novelty of the manuscript is the technological advance in the ability to image single molecule of nascent proteins being translated from mRNA. The authors use nice controls (puromycin treatment, delta NC, NVR etc...) to show that what they are looking at is the result of translation and not random association of protein and mRNA signals. However, this manuscript has a limited scope. These technologies are applicable to viral vector system and not applicable to full length replication competent viruses.

There are issues with writing and figure legends. The figure legends are not clearly described making it difficult to completely understand what is being said. For example, in Figure 4, it is not clear What do open, and closed circles represent. It is indicated that non-translating RNA, is it non-translating FL RNA? IN figure 5 legends, it is stated that "In zooms, the yellow and white arrows show the translating and translating RNA associated with the ER, respectively" Should this be "translating and non-translating RNA?" Similar problems were observed in other figure legends. These need to be corrected.

Re: JCB manuscript #202405075

Dear Editor,

We are grateful to editors Jens Lykke-Andersen and Dan Simon for their encouragement to resubmit our revised article to JCB: "*Given the interest in the topic, we would be open to resubmission to JCB ...*" and "*We would be happy to discuss the reviewer comments further*".

We thank the reviewers for their helpful comments. In response to the reviewers' comments, we have performed several new experiments:

- a new MLV-gGag- Δ NC construct was constructed and added in Fig. 5 and Fig. 6
- a new set of data with new drug (Leptomycin B) was included in new panel C of Fig. 6
- two new Figures were provided, such as FigS4 (Cycloheximide assay) and FigS5 (FL RNA content in virions by using RT-qPCR)

We believe that the additional data have significantly strengthened and improved the manuscript.

We have addressed the reviewers' comments as follows:

Reviewer #1

This study significantly contributes to our understanding of retroviral RNA fate by redefining the translation locations. It will likely interest a broad range of researchers. I have a few suggestions and comments to strengthen the manuscript.

We thank the reviewer for acknowledging the significance of our findings.

Major comments.

- Figure 1: Did the authors try using another translation inhibitor, such as cycloheximide? Cycloheximide stalls ribosomes on mRNA by binding to the ribosome E site and interfering with polypeptide elongation, so the proportion of dual-colored RNA should not decrease with this inhibitor.

Yes, we treated cells expressing MLV-WT with cycloheximide (100 ug/ml) for 15 min, and as expected the percentage of colocalization did not decrease. The data are provided in new Fig. S4

- Figure 2: The authors suggest that the MLV Δ NC construct can be used as a reference to study MLV translation. While I mostly agree, could they measure the amount of gRNA present in viral particles, given that figure S1 shows MLV Δ NC can still form virion progeny (empty or full)? This would mean that Gag Δ NC could still interact with viral RNA.

We quantified FL RNA in cells and in virions by using RT-qPCR. Cells were transfected with the untagged MLV molecular clones (WT or \$\Delta\$ NC) to increase virion release and to facilitate RT and PCR reactions, since the 24XMS2 hairpins interfered with RT and polymerase elongation. Untagged MLV clones allow qPCR amplification of GAPDH mRNA and FL RNA from the same oligodT-RT reaction, normalizing RNA samples extracted from cells.

\$\Delta\$ NC particles lack detectable viral FL RNA (see new Fig.S5), a result that is consistent with previous studies with point mutations or partial deletion (\$\Delta\$ 16-23) of the NC domain of Gag (Gorelick et al, 1988; Meric & Goff 1989; Muriaux et al,2004).

In conclusion, \$\Delta\$ NC Gag does not associate with FL RNA and produces "empty" particles.

- Figure 3: What is the translation rate on MLV WT RNA? Is it different from MLV Δ NC? Could the NC domain influence this rate?

We have performed and analyzed WT videos and found an elongation rate for MLV-WT of 20 AA/sec (see below), which is similar to that of MLV- Δ NC (22 AA/sec). These results do not support a role for NC domain in the translation process. For the sake of homogeneity, we have not included these data in the manuscript where all data were obtained with MLV- Δ NC.

- Figure 5: Did the authors build a mutant where the main Gag AUG initiation codon is substituted to inhibit its translation and favor gGag translation? In this case, do they only observe translation at the ER?

Thanks to the Reviewer for their suggestion. We have deleted the Gag AUG initiation codon in the MLV- Δ NC construct (called MLV-gGag- Δ NC) and its translation was analyzed at the ER (see revised Fig. 5) and at the PM by using TIRFM (see revised Fig.6 A-B). These results showed that the translation sites observed at the ER correspond to gGag synthesis (and not Gag) and the translation sites at/near the PM correspond to Gag synthesis (and not gGag).

- The authors concluded that MLV translation can occur at the PM or cell periphery but they did not discuss the coexistence of two pools of RNA (for translation and packaging) in MLV. Do they think PM-translating MLV mRNAs are the ones to be encapsidated? This would challenge the current understanding of MLV RNA pools and provide a compelling conclusion to the study.

In a previous study, we reported that leptomycin B (LMB) interfered with specific packaging of FL RNA, sequestering the packable FL RNA pool in the nucleus (Mougel et al., 2020). To investigate the coexistence of two functionally distinct pools of MLV FL RNA, we performed TIRFM analysis of cells treated with LMB and determined the proportion of signals colocalized at/near the PM (new Figure 6C). We found that LMB had no effect on the level of local translation, i.e. LMB had no effect on the fate of the FL RNA pool visualized near/at the PM. Consistent with our previous study, the translating FL RNAs present at the PM are exclusively dedicated to translation, confirming the existence of two FL RNA pools with functional separation. The cytoplasmic fate of FL RNA is probably pre-programmed by nuclear export factors that interact with FL RNA.

Reviewer #2

Although complex imaging approaches were used, this study adds little to our current understanding of translation or MLV biology.

To our knowledge, the MLV Gag and gGag syntheses had never been visualized and quantified directly inside cells. Our work provides the first insight into the proportion of FL RNA that is translated, as well as the efficiency and localization of MLV translation.

Major issues:

1. The authors claim that less than 20% of the unspliced RNA is being translated. This is likely to be a significant underestimation because of their experimental design, which only allows visualization of a small portion of the gag gene.

We agree that there is an underestimation, but very little because nascent peptides are detectable long before harboring the 24 motifs of ST. In fact, some studies have been done with only 5xST (S Boersma et al., 2020). In addition, considering the distance (1.1kb) between ribosomes, there are few ribosomes upstream of the ST tag sequence. Experiments performed with an MLV construct containing 12XST showed similar translation rates that obtained with the 24XST version (see below):

The authors could validate their measurement using a standard method to measure the percentage of MLV RNA in polysomes. Without such validation, their conclusion is unsupported.

We believe that the SunTag+MS2 strategy has been well established for the past 10 years and is on the way to becoming a standard method, as traditional polysome profiling approaches. The validity of the method is well supported by the use of translation inhibitor (Puro). To confirm that the colocalized signals correspond to translation sites, we performed experiments with cells treated with a second translation inhibitor, the Cycloheximide (new Fig. S4).

Polysome profiling approach may be weakly suited for MLV mRNA because it has low ribosome coverage, which can result in sparse profiles with significant differences between replicates. MLV FL RNA is very long and highly structured and is prone to fragmentation during sample preparation and degradation by RNases, introducing potential bias (Sugimoto, Y. & Ratcliffe, P.J., 2022; Wang, Q. & Mao, Y., 2023; Chassé H et al., 2017).

2. The authors examined MLV RNA location in five compartments and concluded that RNA is translated in the ER and near the plasma membrane but not in the Golgi, lysosome, or late endosomes. From this, they conclude "translation is regulated in both space and time." Whole-cell imaging showed that MLV RNA is present throughout the cell and there is a large cell-to-cell variation in translation rate (up to 8-fold according to Fig. 1 and 2). To understand whether translation is regulated in space, the comparison should be between RNAs in one of these compartments versus elsewhere in the same cell. The comparison as it currently stands does not tell the whole story.

Now, we calculated the % of translating RNA for each cell and compartment as follows:

$$\% \text{translating RNA} = \frac{\# \text{ colocalized dots in compartment}}{\# \text{ red dots in compartment}}$$

This graph (as Fig. 4) shows that translation is spatially regulated: none at the Golgi, very few at the LE and ER (3-2%) and 23-16% for ER and PM, respectively, while non-translating FL RNAs were almost equally distributed between ER, LE, Lyso and PM (Fig 4 panel C).

3. The significant presence of colocalized RNA and Gag signals (one-third of total detected signals) after puromycin treatment (Fig. 6) suggests these are not translating RNA but Gag:RNA complexes. This very high background is unexpected and should be investigated.

We agree that the background after puromycin treatment in TIRFM images (Fig. 6) was high, but this background did not correspond to Gag-RNA complexes because:

- Gag- Δ NC lacks the ability to interact with FL RNA, as shown by the undetectable level of FL RNA measured by RT-qPCR in the released particles (new Fig. S5), and as documented in the literature (Gorelick et al, 1988; Meric & Goff 1989, Muriaux et al,2004; Li et al, 2014).
- Leptomycin B treatment of cells has no effect on translation level at/near the PM (New Fig. 6C), confirming that colocalized signals imaged by TIRFM are not assembly events, since LMB sequesters the packable FL RNA pool in the nucleus and subsequently its specific packaging into virions (Mougel et al., 2020).

A plausible explanation could be the difference in mesoscale fluidization between cytoplasm and PM. While polysome collapse induced by puromycin treatment increases fluidization of cytoplasm {Xie, 2024 #4552}, membrane tethering or reduced mobility of PM-associated molecules could reduce the diffusivity of the fallen peptides (+puro) and thus its distance from the mRNA, resulting in a higher background of colocalized signals (10%) than that previously observed (3%) inside the cell.

Reviewer #3

The main novelty of the manuscript is the technological advance in the ability to image single molecule of nascent proteins being translated from mRNA. The authors use nice controls (puromycin treatment, delta NC, NVR etc...) to show that what they are looking at is the result of translation and not random association of protein and mRNA signals. However, this manuscript has a limited scope. These technologies are applicable to viral vector system and not applicable to full length replication competent viruses.

We thank the reviewer for acknowledging the novelty and the reliability of our findings.

Aware of the limits of imaging approaches, we inserted tags within the entire molecular clone of MLV, which was still capable of forming and releasing some viral particles.

In general, the criticism of the use of vector systems could be applied to countless publications, but we are convinced that these approaches, just like in vitro studies, contribute greatly to our understanding of living systems.

December 14, 2024

RE: JCB Manuscript #202405075R-A

Marylene Mougel
French National Centre for Scientific Research

Dear Dr. Mougel,

Thank you for submitting your revised manuscript entitled "How and where is the unspliced retroviral genomic RNA translated in the host cell." We would be happy to publish your paper in JCB pending final revisions necessary to meet our formatting guidelines (see details below). Please also address Reviewer #2's remaining comment by adding a sentence on the possible underestimation of the percentage of translated RNAs.

A. MANUSCRIPT ORGANIZATION AND FORMATTING:

1) Text limits: Character count for Articles is < 40,000, not including spaces. Count includes title page, abstract, introduction, results, discussion, and acknowledgments. Count does not include materials and methods, figure legends, references, tables, or supplemental legends.

2) Figure formatting: Articles may have up to 10 main text figures. Scale bars must be present on all microscopy images, including inset magnifications. Molecular weight or nucleic acid size markers must be included on all gel electrophoresis. Also, please avoid pairing red and green for images and graphs to ensure legibility for color-blind readers. If red and green are paired for images, please ensure that the particular red and green hues used in micrographs are distinctive with any of the colorblind types. If not, please modify colors accordingly or provide separate images of the individual channels.

3) Statistical analysis: Error bars on graphic representations of numerical data must be clearly described in the figure legend. The number of independent data points (n) represented in a graph must be indicated in the legend. Please, indicate whether 'n' refers to technical or biological replicates (i.e. number of analyzed cells, samples or animals, number of independent experiments). If independent experiments with multiple biological replicates have been performed, we recommend using distribution-reproducibility SuperPlots (please see Lord et al., JCB 2020) to better display the distribution of the entire dataset, and report statistics (such as means, error bars, and P values) that address the reproducibility of the findings.

Statistical methods should be explained in full in the materials and methods. For figures presenting pooled data the statistical measure should be defined in the figure legends. Please also be sure to indicate the statistical tests used in each of your experiments (both in the figure legend itself and in a separate methods section) as well as the parameters of the test (for example, if you ran a t-test, please indicate if it was one- or two-sided, etc.). Also, if you used parametric tests, please indicate if the data distribution was tested for normality (and if so, how). If not, you must state something to the effect that "Data distribution was assumed to be normal but this was not formally tested."

4) Abstract and title: The title should not be in the form of a question and needs to clearly convey the main advance. Therefore we suggest the following title: "Translation of unspliced retroviral genomic RNA in the host cell is regulated in both space and time."

The abstract is fine but we recommend simplifying the first sentence to: "Retroviruses carry a genomic intron-containing RNA with a long structured 5'-untranslated region, which acts either as a genome encapsidated in the viral progeny or as an mRNA encoding the key structural protein Gag."

5) Materials and methods: Should be comprehensive and not simply reference a previous publication for details on how an experiment was performed. Please provide full descriptions (at least in brief) in the text for readers who may not have access to referenced manuscripts. The text should not refer to methods "...as previously described." JCB formatting does not allow for supplemental methods. Please move these into the main methods section.

6) For all cell lines, vectors, constructs/cDNAs, etc. - all genetic material: please include database / vendor ID (e.g. Addgene, ATCC, etc.) or if unavailable, please briefly describe their basic genetic features, even if described in other published work or gifted to you by other investigators (and provide references where appropriate). Please be sure to provide the sequences for all

of your oligos: primers, si/shRNA, RNAi, gRNAs, etc. in the materials and methods. You must also indicate in the methods the source, species, and catalog numbers/vendor identifiers (where appropriate) for all of your antibodies, including secondary. If antibodies are not commercial, please add a reference citation if possible.

7) Microscope image acquisition: The following information must be provided about the acquisition and processing of images:

- a. Make and model of microscope
- b. Type, magnification, and numerical aperture of the objective lenses
- c. Temperature
- d. Imaging medium
- e. Fluorochromes
- f. Camera make and model
- g. Acquisition software
- h. Any software used for image processing subsequent to data acquisition. Please include details and types of operations involved (e.g., type of deconvolution, 3D reconstitutions, surface or volume rendering, gamma adjustments, etc.).

8) References: There is no limit to the number of references cited in a manuscript. References should be cited parenthetically in the text by author and year of publication. Abbreviate the names of journals according to PubMed.

9) Supplemental materials: Articles generally may have up to 5 supplemental figures and 10 videos. You currently exceed this limit and while in this case we can give you extra space, please try combine these if possible. Please also note that tables, like figures, should be provided as individual, editable files. A summary of all supplemental material should appear at the end of the Materials and methods section. Please include one brief sentence per item.

10) Video legends: Should describe what is being shown, the cell type or tissue being viewed (including relevant cell treatments, concentration and duration, or transfection), the imaging method (e.g., time-lapse epifluorescence microscopy), what each color represents, how often frames were collected, the frames/second display rate, and the number of any figure that has related video stills or images.

11) eTOC summary: A ~40-50 word summary that describes the context and significance of the findings for a general readership should be included on the title page. The statement should be written in the present tense and refer to the work in the third person. It should begin with "First author name(s) et al..." to match our preferred style.

13) A separate author contribution section is required following the Acknowledgments in all research manuscripts. All authors should be mentioned and designated by their first and middle initials and full surnames. We encourage use of the CRediT nomenclature (<https://casrai.org/credit/>).

14) ORCID IDs: ORCID IDs are unique identifiers allowing researchers to create a record of their various scholarly contributions in a single place. Please note that ORCID IDs are required for all authors. At resubmission of your final files, please be sure to provide your ORCID ID and those of all co-authors.

15) JCB requires authors to submit Source Data used to generate figures containing gels and Western blots with all revised manuscripts. This Source Data consists of fully uncropped and unprocessed images for each gel/blot displayed in the main and supplemental figures. Since your paper includes cropped gel and/or blot images, please be sure to provide one Source Data file for each figure that contains gels and/or blots along with your revised manuscript files. File names for Source Data figures should be alphanumeric without any spaces or special characters (i.e., SourceDataF#, where F# refers to the associated main figure number or SourceDataFS# for those associated with Supplementary figures). The lanes of the gels/blots should be labeled as they are in the associated figure, the place where cropping was applied should be marked (with a box), and molecular weight/size standards should be labeled wherever possible. Source Data files will be directly linked to specific figures in the published article.

16) Journal of Cell Biology now requires a data availability statement for all research article submissions. These statements will be published in the article directly above the Acknowledgments. The statement should address all data underlying the research presented in the manuscript. Please visit the JCB instructions for authors for guidelines and examples of statements at (<https://rupress.org/jcb/pages/editorial-policies#data-availability-statement>).

B. FINAL FILES:

Thank you for your attention to these final processing requirements. Please revise and format the manuscript and upload materials within 7 days. If you need an extension for whatever reason, please let us know and we can work with you to determine a suitable revision period.

Thank you for this interesting contribution, we look forward to publishing your paper in Journal of Cell Biology.

Sincerely,

Jens Lykke-Andersen, PhD
Monitoring Editor
Journal of Cell Biology

Dan Simon, PhD
Scientific Editor
Journal of Cell Biology

Reviewer #1 (Comments to the Authors (Required)):

Thanks to the authors who took into account my comments and suggestions.
The addition of the two mutants (Gag versus glycoGag) really strengthen the manuscript.

Reviewer #2 (Comments to the Authors (Required)):

With added control experiments, this manuscript has improved. One point that remains to be addressed is the percentage of translating RNA measured by the Suntag method. The author reported that only ~20% of the MLV FL RNA is being translated. As Suntag was inserted in the middle of the gene, it is very likely that 20% is an underestimation. The authors have rejected the suggestion that validation should be made using a polysome assay, which was used in multiple retroviral translation publications. At the very least, they should clearly state in the Result Section that the strategy can lead to an underestimation of translating RNA.

Reviewer #3 (Comments to the Authors (Required)):

In this current manuscript the authors investigate the translational activity of MLV viral RNA using dual labelling system of MS2-MCP and ST technologies. These technologies have allowed them to visualize single molecules of RNA undergoing translation in infected cells. Using MLV system, they were able to visualize site of translation of the viral RNA when it is prevented from interacting with NC and hence prevented from being actively engaged in assembly. They have made several interesting observations, including the fact that the site of translation of mRNA includes places near Plasma membrane where assembly is taking place and synthesis of glycoGag at the ER. The experiments are well designed, and several controls are used to ensure what is being scored is RNA being translated and not non-specific association.

In the revised manuscript addition of several controls including the use of another translational inhibitor with different mechanism, quantitation of DeltaNC RNA in the virions, quantitation of elongation rate of WT and DeltaNC RNA etc.... support their hypothesis and further strengthen the manuscript.